# Multicenter international assessment of a SARS-CoV-2 RT-LAMP test for point of care clinical application

Suying Lu[1,2,3], David Duplat[4], Paula Benitez-Bolivar[4], Cielo León[4], Stephany D. Villota[5], Eliana Veloz-Villavicencio[5], Valentina Arévalo[5], Katariina Jaenes[6], Yuxiu Guo[7], Seray Cicek[7], Lucas Robinson[8], Philippos Peidis[1,2,3], Joel D. Pearson[1,2,3], Jim Woodgett[1,9], Tony Mazzulli[2,10], Patricio Ponce[5], Silvia Restrepo[11], John M. González[12], Adriana Bernal[13], Marcela Guevara-Suarez[14], Keith Pardee[6,7,15], Varsovia E. Cevallos[5], Camila González[4], Rod Bremner[1,2,3]*

1 Lunenfeld Tanenbaum Research Institute, Mt Sinai Hospital, Sinai Health System, Toronto, ON, Canada, 2 Department of Laboratory Medicine and Pathobiology, University of Toronto, Toronto, ON, Canada, 3 Department of Ophthalmology and Vision Science, University of Toronto, Toronto, ON, Canada, 4 Centro de Investigaciones en Microbiología y Parasitología Tropical (CIMPAT), Department of Biological Sciences, Universidad de los Andes, Bogotá, Colombia, 5 Centro de Investigación en Enfermedades Infecciosas y Vectoriales (CIREV), Instituto Nacional de Investigación en Salud Pública, Quito, Ecuador, 6 Leslie Dan Faculty of Pharmacy, University of Toronto, Toronto, ON, Canada, 7 LSK Technologies Inc., Kitchener, Canada, 8 Velocity, University of Waterloo, Kitchener, ON, Canada, 9 Department of Medical Biophysics, University of Toronto, Toronto, Canada, 10 Department of Microbiology, Sinai Health System/University Health Network, Toronto, Canada, 11 Department of Food and Chemical Engineering, Universidad de los Andes, Bogotá, Colombia, 12 Grupo de Ciencias Básicas Médicas, School of Medicine, Universidad de los Andes, Bogotá, Colombia, 13 Laboratory of Molecular Interactions of Agricultural Microbes (LIMMA), Department of Biological Sciences, Universidad de Los Andes, Bogotá, Colombia, 14 Applied genomics research group, Vicerrectoría de Investigación y Creación, Universidad de los Andes, Bogotá, Colombia, 15 Department of Mechanical and Industrial Engineering, University of Toronto, Toronto, ON, Canada

* bremner@lunenfeld.ca

**Data Availability Statement:** All relevant data are within the paper and its Supporting Information files.

## Abstract

Continued waves, new variants, and limited vaccine deployment mean that SARS-CoV-2 tests remain vital to constrain the coronavirus disease 2019 (COVID-19) pandemic. Affordable, point-of-care (PoC) tests allow rapid screening in non-medical settings. Reverse-transcription loop-mediated isothermal amplification (RT-LAMP) is an appealing approach. A crucial step is to optimize testing in low/medium resource settings. Here, we optimized RT-LAMP for SARS-CoV-2 and human β-actin, and tested clinical samples in multiple countries. "TTTT" linker primers did not improve performance, and while guanidine hydrochloride, betaine and/or Igepal-CA-630 enhanced detection of synthetic RNA, only the latter two improved direct assays on nasopharygeal samples. With extracted clinical RNA, a 20 min RT-LAMP assay was essentially as sensitive as RT-PCR. With raw Canadian nasopharyngeal samples, sensitivity was 100% (95% CI: 67.6% - 100%) for those with RT-qPCR Ct values $\leq$ 25, and 80% (95% CI: 58.4% - 91.9%) for those with 25 < Ct $\leq$ 27.2. Highly infectious, high titer cases were also detected in Colombian and Ecuadorian labs. We further demonstrate the utility of replacing thermocyclers with a portable PoC device (FluoroPLUM). These combined PoC molecular and hardware tools may help to limit community transmission of SARS-CoV-2.

**Funding:** Funding: Canada's International Development Research Centre (RB, CG and KP, grant number: 109547-001) through the COVID-19 May 2020 Rapid Research Funding Opportunity, the Krembil Foundation (RB, grant number: Not applicable), and the University of Toronto COVID-19 Action Initiative 2020 (KP, grant number: Not applicable). The funders had no role in study design, data collection and analysis, decision to publish, or preparation of the manuscript.

**Competing interests:** I have read the journal's policy and the authors of this manuscript have the following competing interests: Y.G., S.C., and K.P. are co-inventors of the PLUM reader and co-founders of LSK Technologies, Inc. No other commercial declarations are relevant to this study. This does not alter our adherence to PLOS ONE policies on sharing data and materials.

# Introduction

With continuing waves of coronavirus disease 2019 (COVID-19) around the world, there has been sustained focus on testing to mitigate and suppress spread of the disease [1]. Limited vaccination and the emergence of new variants [2], most recently Omicron [3], exacerbate recurrent viral surges. Viral shedding in COVID-19 patients peaks on or before symptom onset, and contact tracing and quarantine should be done at a crucial temporal window 2 to 3 days before demonstration of symptoms [4,5], although the exact timing to obtain reliable results is debated [6,7]. Although current gold-standard quantitative real-time polymerase chain reaction (qPCR) assays have sensitive analytical limits of detection (LoD), they are generally performed in sophisticated detection centers with high cost and long turnaround times [8]. Computer modelling studies based on the pattern of viral load kinetics show that effective community control of transmission depends more on testing frequency and shorter turnaround times, than analytical LoD [8]. Further, reducing the barriers to testing may also provide significant benefit in settings where point-of-need applications are time sensitive and infrastructure is limited (e.g. school testing and travel). Viral load correlates negatively with cycle threshold (Ct) values and positively with infectivity [9]. A few reports suggested that COVID-19 patients with Ct values ≤ 25 are more likely to be infectious while patients with Ct values above 33–34 are not contagious [10–12]. Modelling further shows that routine testing substantially reduces risk of COVID-19 outbreaks in high-risk healthcare environments, and may need to be as frequent as twice weekly [13]. Effective COVID-19 containment demands point-of-care (PoC) tests with short turnaround time, low cost and high accessibility [14]. Indeed, many rapid PoC antigen and molecular-based tests for diagnosis of SARS-CoV-2 infection have been developed with a wide range of detection sensitivity and overall high specificity [14]. Some of these tests are approved by regulatory agencies and commercially available [14]. However, the high cost of the rapid antigen tests and the requirement of specialized automated instruments for the molecular-based tests [14] limits accessibility to broad communities.

Reverse transcription loop-mediated isothermal amplification (RT-LAMP) can be performed in a low-resource setting by merely heating the samples and reagents in a single reaction tube at one constant temperature, and diagnosis is available within 30 minutes [15]. RT-LAMP has clear advantages over RT-PCR as a PoC test, and it has been applied to diagnose several viral diseases, such as Severe Acute Respiratory Syndrome (SARS) and Middle East Respiratory Syndrome (MERS), among others [15]. The scientific community has applied RT-LAMP to detect SARS-CoV-2 in different kinds of samples, using different primers and experimental readouts. In most of those studies, viral detection required purified RNA or sample treatment, and generated variable detection efficiencies [15]. Among seven isothermal tests with Emergency Use Approval (EUA), the LoDs vary up to 50-fold, and are much less sensitive than those of RT-PCR [16]. Recent studies have utilized RT-LAMP on direct patient samples without any RNA purification [17–21], proving feasibility of this approach. However, these studies did not examine assay robustness in different settings, particularly in low resource countries where reagent availability can be a major roadblock. Here, we set out to develop an optimized RT-LAMP assay and assess feasibility and robustness in different low resource countries.

Using a commercially available RT-LAMP kit, we performed systematic primer optimization, and further improved sensitivity with primer multiplexing, and various additives. Using purified RNA as the template, the optimized RT-LAMP assay has similar sensitivity and specificity to commercial RT-PCR kits used widely in the clinic. Direct RT-LAMP with raw clinical samples was less efficient, but detected high titer samples from patients predicted to be

infectious with high specificity and sensitivity. At a stringent cutoff of 100% specificity (no false positives) as the United States Food and Drug Administration (FDA) recommended [22], labs using the RT-LAMP assay in Canada, Colombia and Ecuador displayed a range of sensitivities, but each could detect highly infectious disease. Finally, using a PoC instrument that enables de-centralized deployment of the RT-LAMP assay, we describe the application of this test on raw unpurified samples. This direct RT-LAMP strategy reduces the barrier to establishing testing capacity by overcoming the need for laboratory infrastructure for RNA extraction or specialized thermocycling and optical monitoring equipment. This detection method has potential as a PoC test to screen individuals with high viral loads and mitigate viral transmission.

## Materials and methods

### Oligos

All oligos (Table 1) were ordered from IDT and dissolved with DNase/RNase-free water at 100μM concentration. The purification method for F3, B3, LF and LB was standard desalting, and the purification method for FIP and BIP was HPLC. Oligos for each primer set were combined to make a 10X mix based on required concentrations.

### Control SARS-CoV-2 RNA

Synthetic SARS-CoV-2 viral RNA sequences were ordered from Twist Bioscience (Cat. No. 102019, $1X10^6$ RNA copies/μl), and they were non-overlapping fragments of the genome appropriate for each set of primers. The viral RNAs were diluted with DNase/RNase-free water accordingly based on the need of experiments.

### Clinical nasopharyngeal (NP) samples

Canadian samples: 30 SARS-CoV-2 positive and 36 negative heated-inactivated clinical NP samples in Universal Transport Medium (UTM) were provided by the Microbiology Department of Mount Sinai Hospital/University Health Network in Toronto, Canada. These samples were collected from January to July 2020. The sample size was determined based on FDA recommendation regarding development of molecular diagnostic test for SARS-CoV-2 [22]. The samples were kept at -80˚C in a Viral Tissue Culture (VTC) laboratory in the Lunenfeld-Tanenbaum Research Institute (LTRI), and all the experiments related to these samples were performed in the VTC lab. These samples were surplus diagnostic materials that were analyzed anonymously, and no specific approval from Research Ethics Board (REB) was required. The clinical information regarding these samples was not known.

Colombian samples: Two batches of clinical NP swab samples were chosen from the samples collected previously for the Uniandes COVIDA project, and were collected between February 16th and March 29th of 2021. Batch 1: 134 positive and 50 negative samples were re-evaluated by qRT-PCR with freshly extracted RNA to confirm SARS-CoV-2 status and sample integrity. With the exclusion of the samples with Ct > 38 for Orf1ab, N gene and RNase P, 41 negative samples and 118 positive samples with Ct values for SARS-CoV-2 Orf1ab from 15 to 36.4 were selected to optimize direct RT-LAMP. Batch 2: 120 positive and 120 negative samples were randomly chosen, and re-evaluated to confirm SARS-CoV-2 condition and sample quality. With the exclusion of the samples with Ct > 38 for Orf1ab, N gene and RNase P, 88 positive and 120 negative samples were selected to test the direct RT-LAMP assay.

**Table 1. Primer sets that were optimized for RT-LAMP.**

| Primer name | Primer sequence | Sequence targeted | References | Phase |
|---|---|---|---|---|
| ORF1a-C-F3 | CTGCACCTCATGGTCATGTT | 498–517 in GeneBank: MT007544.1 | Zhang Y et al. [23] | 1[a] |
| ORF1a-C-B3 | GATCAGTGCCAAGCTCGTC | 704–722 in GeneBank: MT007544.1 | | |
| ORF1a-C-LF | ACCACTACGACCGTACTGAAT | ORF1a of SARS-CoV-2 | | |
| ORF1a-C-LB | TTCGTAAGAACGGTAATAAAGGAGC | | | |
| ORF1a-C-FIP | GAGGGACAAGGACACCAAGTGTGGTAGCAGAACTCGAAGGC | | | |
| ORF1a-C-BIP | CCAGTGGCTTACCGCAAGGTTTTAGATCGGCGCCGTAAC | | | |
| ORF1a-C-TFIP | GAGGGACAAGGACACCAAGTG**TTTT**TGGTAGCAGAACTCGAAGGC | | | |
| ORF1a-C-TBIP | CCAGTGGCTTACCGCAAGGTT**TTTT**TTAGATCGGCGCCGTAAC | | | |
| As1_F3 | CGGTGGACAAATTGTCAC | 2245–2262 in GeneBank: MT007544.1 | Rabe BA et al.[24] | 1[a] |
| As1_B3 | CTTCTCTGGATTTAACACACTT | 2420–2441 in GeneBank: MT007544.1 | | |
| As1_LF | TTACAAGCTTAAAGAATGTCTGAACACT | ORF1a of SARS-CoV-2 | | |
| As1_LB | TTGAATTTAGGTGAAACATTTGTCACG | | | |
| As1e_FIP | TCAGCACACAAAGCCAAAAATTTAT**TTTT**CTGTGCAAAGGAAATTAAGGAG | | | |
| As1e_BIP | TATTGGTGGAGCTAAACTTAAAGCC**TTTT**CTGTACAATCCCTTTGAGTG | | | |
| ORF1a-F3 | TCCAGATGAGGATGAAGAAGA | 3043–3063 in GeneBank: MT007544.1 | Lamb LE et al.[25] | 1[a] 2[b] 3[c] 4[d] 5[e] |
| ORF1a-B3 | AGTCTGAACAACTGGTGTAAG | 3311–3331 in GeneBank: MT007544.1 | | |
| ORF1a-LF | CTCATATTGAGTTGATGGCTCA | ORF1a of SARS-CoV-2 | | |
| ORF1a-LB | ACAAACTGTTGGTCAACAAGAC | | | |
| ORF1a-FIP | AGAGCAGCAGAAGTGGCACAGGTGATTGTGAAGAAGAAGAG | | | |
| ORF1a-BIP | TCAACCTGAAGAAGAGCAAGAACTGATTGTCCTCACTGCC | | | |
| ORF1a-TFIP | AGAGCAGCAGAAGTGGCAC**TTTT**AGGTGATTGTGAAGAAGAAGAG | | | |
| ORF1a-TBIP | TCAACCTGAAGAAGAGCAAGAA**TTTT**CTGATTGTCCTCACTGCC | | | |
| GeneE1-F3 | TGAGTACGAACTTATGTACTCAT | 26232–26254 in GeneBank: MT007544.1 | Zhang Y et al.[26] | 1[a] 2[b] 3[c] 4[d] 5[e] |
| GeneE1-B3 | TTCAGATTTTTAACACGAGAGT | 26420–26441 in GeneBank: MT007544.1 | | |
| GeneE1-LF | CGCTATTAACTATTAACG | Gene E of SARS-CoV-2 | | |
| GeneE1-LB | GCGCTTCGATTGTGTGCGT | | | |
| GeneE1-FIP | ACCACGAAAGCAAGAAAAAGAAGTTCGTTTCGGAAGAGACAG | | | |
| GeneE1-BIP | TTGCTAGTTACACTAGCCATCCTTAGGTTTTACAAGACTCACGT | | | |
| GeneE1-TFIP | ACCACGAAAGCAAGAAAAAGAAG**TTTT**TTCGTTTCGGAAGAGACAG | | | |
| GeneE1-TBIP | TTGCTAGTTACACTAGCCATCCTTA**TTTT**GGTTTTACAAGACTCACGT | | | |
| GeneN-A-F3 | TGGCTACTACCGAAGAGCT | 28525–28543 in GeneBank: MT007544.1 | Zhang Y et al. [23] | 1[a] |
| GeneN-A-B3 | TGCAGCATTGTTAGCAGGAT | 28722–28741 in GeneBank: MT007544.1 | | |
| GeneN-A-LF | GGACTGAGATCTTTCATTTTACCGT | Gene N of SARS-CoV-2 | | |
| GeneN-A-LB | ACTGAGGGAGCCTTGAATACA | | | |
| GeneN-A-FIP | TCTGGCCCAGTTCCTAGGTAGTCCAGACGAATTCGTGGTGG | | | |
| GeneN-A-BIP | AGACGGCATCATATGGGTTGCACGGGTGCCAATGTGATCT | | | |
| GeneN-A-TFIP | TCTGGCCCAGTTCCTAGGTAGT**TTTT**CCAGACGAATTCGTGGTGG | | | |
| GeneN-A-TBIP | AGACGGCATCATATGGGTTGCA**TTTT**CGGGTGCCAATGTGATCT | | | |

(*Continued*)

**Table 1.** (Continued)

| Primer name | Primer sequence | Sequence targeted | References | Phase |
|---|---|---|---|---|
| **N-gene-F3** | AACACAAGCTTTCGGCAG | 29083–29100 in GeneBank: MT007544.1 | Broughton JP *et al.* [27] | 1[a] 2[b] |
| **N-gene-B3** | GAAATTTGGATCTTTGTCATCC | 29290–29311 in GeneBank: MT007544.1 | | |
| **N-gene-LF** | TTCCTTGTCTGATTAGTTC | Gene N of SARS-CoV-2 | | |
| **N-gene-LB** | ACCTTCGGGAACGTGGTT | | | |
| **N-gene-FIP** | TGCGGCCAATGTTTGTAATCAGCCAAGGAAATTTTGGGGAC | | | |
| **N-gene-BIP** | CGCATTGGCATGGAAGTCACTTTGATGGCACCTGTGTAG | | | |
| **N-gene-TFIP** | CGCATTGGCATGGAAGTCAC**TTTT**TTTGATGGCACCTGTGTAG | | | |
| **N-gene-TBIP** | TGCGGCCAATGTTTGTAATCAG**TTTT**CCAAGGAAATTTTGGGGAC | | | |
| **Gene N2-F3** | ACCAGGAACTAATCAGACAAG | 29136–29156 in GeneBank: MT007544.1 | Zhang Y *et al.*[26] | 1[a] 2[b] 3[c] |
| **Gene N2-B3** | GACTTGATCTTTGAAATTTGGATCT | 29299–29323 in GeneBank: MT007544.1 | | |
| **Gene N2-LF** | GGGGGCAAATTGTGCAATTTG | Gene N of SARS-CoV-2 | | |
| **Gene N2-LB** | CTTCGGGAACGTGGTTGACC | | | |
| **Gene N2-FIP** | TTCCGAAGAACGCTGAAGCG–GAACTGATTACAAACATTGGCC | | | |
| **Gene N2-BIP** | CGCATTGGCATGGAAGTCAC–AATTTGATGGCACCTGTGTA | | | |
| **Gene N2-TFIP** | TTCCGAAGAACGCTGAAGCG**TTTT**GAACTGATTACAAACATTGGCC | | | |
| **Gene N2-TBIP** | CGCATTGGCATGGAAGTCAC**TTTT**AATTTGATGGCACCTGTGTA | | | |
| **ACTB-F3** | AGTACCCCATCGAGCACG | 287–304 in NM_001101.5 | Zhang Y *et al.*[26] | 1[a] 2[b] 4[d] 5[e] |
| **ACTB-B3** | AGCCTGGATAGCAACGTACA | 479–498 in NM_001101.5 | | |
| **ACTB-LF** | TGTGGTGCCAGATTTTCTCCA | Human ACTB mRNA | | |
| **ACTB-LB** | CGAGAAGATGACCCAGATCATGT | | | |
| **ACTB-FIP** | GAGCCACACGCAGCTCATTGTATCACCAACTGGGACGACA | | | |
| **ACTB-BIP** | CTGAACCCCAAGGCCAACCGGCTGGGGTGTTGAAGGTC | | | |
| **ACTB-TFIP** | GAGCCACACGCAGCTCATTGTA**TTTT**TCACCAACTGGGACGACA | | | |
| **ACTB-TBIP** | CTGAACCCCAAGGCCAACCG**TTTT**GCTGGGGTGTTGAAGGTC | | | |

Summary of the above primer sets in optimization phases.

[a]Phase 1: Primer screening with 30 copies of synthetic SARS-CoV-2 RNA (all primer sets).

[b]Phase 2: Defining LoDs with 30, 60, 120 and 240 copies of synthetic SARS-CoV-2 RNA or 0.01, 0.05, 0.25 and 1.25ng of human RNA (ORF1a, E1, N-gene N2, and ACTB).

[c]Phase 3: Maximizing sensitivity by primer multiplexing and supplementation GuHCl and/or Betaine with 15 copies of SARS-CoV-2 synthetic RNA (ORF1a and E1).

[d]Phase 4: Detecting SARS-CoV-2 with extracted RNA from clinical NP samples by multiplexing ORF1a and E1 and supplementing GuHCl and Betaine (ORF1a and E1).

[e]Phase 5: Detecting SARS-CoV-2 with raw clinical NP samples by multiplexing ORF1a and E1 and supplementing Betaine and Igepal CA-630 (ORF1a and E1).

Ecuadorian samples: 21 positive and 21 negative NP swab samples were collected from February to August 2021 in Quito Ecuador. These samples were used to test the optimized RT-LAMP with extracted RNA and raw samples.

## RNA extraction from clinical NP samples

Canadian samples: RNA extraction from clinical NP samples was carried out with miRNeasy Mini Kit (Qiagen, Cat. No. 217004) according to the kit instructions. For all the SARS-CoV-2 positive or negative NP samples, 50μl was aliquoted for RNA extraction, and the extracted RNA was eluted out with 50μl DNase/RNase free water.

Colombian samples: Batch 1: RNA extraction was performed with Quick-RNA viral kit (Zymo, Cat. No. R1035-E). 100μl of sample was applied for extraction, and RNA was eluted in

50μl RNAse free water. Batch 2: Extraction was performed using the Nextractor NX-48S (Genolution), an automated system for rapid DNA/RNA isolation, 200 μl of sample was applied for extraction. RNAse free water was added to the eluted until it reached the 200 μL.

Ecuadorian samples: RNA extraction was performed with ExtractMe viral RNA kit (Blirt, Cat. No. EM39) following manufacturer's instructions. 100μl of sample was used for extraction, and RNA was eluted out with 30μl RNase free water.

## Generation of contrived positive NP samples

To better evaluate the detection sensitivity of the maximized RT-LAMP with raw NP samples, 12 SARS-CoV-2 positive clinical NP samples from the Canadian cohort were diluted with 12 negative clinical NP samples to create 56 contrived positive NP samples with predicted Ct values between 21.0 and 31.0.

## Optimization of RT-LAMP

RT-LAMP was optimized with SARS-CoV-2 RNA and raw NP samples without RNA extraction respectively. The optimization for detecting SARS-CoV-2 with extracted RNA was mainly based on LoD for each assay condition with different copy numbers of SARS-CoV-2 RNA. With each copy number, 10 replicates were tested, and LoD was defined as the lowest copy number of SARS-CoV-2 RNA detected in 100% (10/10) of replicates. The optimization for detecting SARS-CoV-2 in raw NP samples was carried out with 30 positive and 36 negative clinical NP samples to test each assay condition, and receiver operating characteristic (ROC) curve analysis was applied to evaluate sensitivity and specificity.

Before setting up the RT-LAMP experiments, bench surface, racks and pipettes were decontaminated with 10% bleach and 70% alcohol. The main reagents for RT-LAMP were WarmStart colorimetric LAMP 2X Master Mix (NEB, Cat. No. M1800L) and 5mM STYO 9 Green Fluorescent Nucleic Acid Stain (Life technologies, Cat. No. S34854). Other reagents for the optimization were GuHCl (Sigma-Aldrich, Cat. No. G3272-25G), 5M betaine (Sigma-Aldrich, Cat. No. B0300-1VL) and Igepal CA-630 (Sigma-Aldrich, Cat. No. I8896). The volume for each RT-LAMP reaction was 10μl, including 5μl WarmStart colorimetric LAMP 2X Master Mix, 1μl 10X primer set stock and 1μl template. The remaining volume was filled with H2O or supplements. The RT-LAMP reactions were set up on ice, and were carried out with 384-well plates (ThermoFisher, Cat. No. 4309849) at 65˚C using CFX 384 Real-Time System (BIO-RAD) operated with Bio-Rad CFX manager 3.1. The plate reading was set for SYBR green reading, and read every 30 seconds, total 120 reads. At the end of the experiments, color images of the 384-well plates were scanned with a Canon photocopier because the commercial RT-LAMP kit is designed to produce a change in solution color from pink to yellow with the presence of amplification. In the experiments performed with FluoroPLUM (LSK Technologies Inc., Cat. No. SPF), 96-well plates (Luna Nanotech, Cat. No. MPPCRN-NH96W) were used. The optimized RT-LAMP recipes for various conditions were in Table 2. The optimized RT-LAMP assays were evaluated in Colombian and Ecuadorian laboratories with CFX96™ Real-Time System (BIO-RAD) using 96-well plates (BIO-RAD, Cat. No. HSP9601).

## RT-qPCR

Canadian samples: Before performing experiments, bench surface, racks and pipettes were cleaned with 10% bleach, 70% alcohol and DNAZap (Thermofisher, Cat. No. AM9890). BGI Real-Time Fluorescent RT-PCR Kit for Detecting SARS-CoV-2 (Cat. No. MFG030018) was applied according to the kit instructions with some modifications. RT-PCR reagents and RNA samples were thawed and kept on ice. For each 10μl RT-PCR reaction, 6.17μl SARS-CoV-2

**Table 2. Optimized RT-LAMP conditions.**

| Templates | Primer concentrations | Supplements[a] | SYTO 9[a] | Instruments |
|---|---|---|---|---|
| Extracted RNA for SARS-CoV-2 | ORF1a: F3B3(0.2µM)/FIPBIP(3.2µM)/LFLB(0.4µM)<br>Gene E1: F3B3(0.2µM)/FIPBIP(3.2µM)/LFLB(0.4µM) | 40mM GuHCl<br>0.5M betaine | 1µM | CFX 384 Real-Time System |
| Extracted RNA for ACTB | ACTB:<br>F3B3(0.05µM)/FIPBIP(0.4µM)/LFLB(0.1µM) | 40mM GuHCl<br>0.5M betaine | 1µM | CFX 384 Real-Time System |
| Raw NP samples for SARS-CoV-2 | ORF1a: F3B3(0.2µM)/FIPBIP(3.2µM)/LFLB(0.4µM)<br>Gene E1: F3B3(0.2µM)/FIPBIP(3.2µM)/LFLB(0.4µM) | 0.5M betaine<br>0.25% Igepal CA-630 | 1µM | CFX 384 Real-Time System |
| Raw NP samples for ACTB | ACTB:<br>F3B3(0.05µM)/FIPBIP(0.4µM)/LFLB(0.1µM) | 0.5M betaine<br>0.25% Igepal CA-630 | 1µM | CFX 384 Real-Time System |
| Raw NP samples for SARS-CoV-2 | ORF1a: F3B3(0.2µM)/FIPBIP(3.2µM)/LFLB(0.4µM)<br>Gene E1: F3B3(0.2µM)/FIPBIP(3.2µM)/LFLB(0.4µM) | 0.5M betaine<br>0.25% Igepal CA-630 | 10µM | FluoroPLUM |
| Raw NP samples for ACTB | ACTB:<br>F3B3(0.05µM)/FIPBIP(0.4µM)/LFLB(0.1µM) | 0.5M betaine<br>0.25% Igepal CA-630 | 10µM | FluoroPLUM |

[a]Appropriate concentrations for experiments were prepared with DNase/RNase-free water.

Reaction Mix, 0.5µl SARS-CoV-2 Enzyme Mix, 2.33µl DNase/RNase-free water and 1µl template was loaded to a well of 384-well plate (BIO-RAD, Cat. No. HSP3805). The RT-PCR reaction was carried out with CFX 384 Real-Time System (Bio-Rad Laboratories) operated with Bio-Rad CFX Manager 3.1, and the plate reading was set as defined by the kit instructions for all channel reading. A sample was defined as SARS-CoV-2 positive if the Ct for ORF1ab was < 37.0. A sample was defined as ACTB positive if the Ct for ACTB was < 35.0.

Colombian samples: U-TOP Seasun RT-PCR kit was used to detect viral RNA and human RNase P gene following the kit instructions. A sample was defined as SARS-CoV-2 positive if Ct value for Orf1ab and/or N gene was ≤ 38. The cutoff for RNase P was Ct ≤ 38 as well.

Ecuadorian samples: The qRT-PCR was performed with an in-house assay with targets in the N and E genes based on the following protocols [28,29]. The SuperScript™ III Platinum™ One-Step RT-qPCR Kit (Invitrogen, Cat. No. 12574026) was used to detect specific targets. The cutoff Ct value for the gene E was 30, and 35 for the gene N and human ACTB.

## Evaluation of RT-LAMP performance with receiver operating characteristic (ROC) curve analysis

For all the positive and negative clinical NP samples confirmed by BGI RT-PCR kit, RT-LAMP TTR (time to results) or slope$_{20-40}$ was plotted in functions of the true positive rate (Sensitivity) and the false positive rate (1-Specificity) for ROC curve analysis using MedCalc software [30]. The area under the ROC curve (AUC) and cut-off TTR or slope$_{20-40}$ at which the true positive plus true negative rate is highest was calculated. More stringent cutoffs were used in some cases, as indicated in the text, to achieve 100% specificity [22].

## Results

### Phase 1: Screening primers at a low template copy number

The RT-LAMP reagent used in this study is WarmStart®Colorimetric LAMP 2X Master Mix (NEB, Cat. No. M1800L), and employed four core primers: FIP (forward inner primer), BIP (backward inner primer), F3 (forward primer), B3 (backward primer) to amplify the target region, and two loop primers, LF (loop forward) and LB (loop backward), to enhance reaction speed (Fig 1A). In LAMP reactions, non-specific amplification is common due to *cis* and *trans* priming among the six primers [31]. Robust performance of RT-LAMP requires thorough optimization of the six primers over a wide range of concentrations [32]. Including a "TTTT"

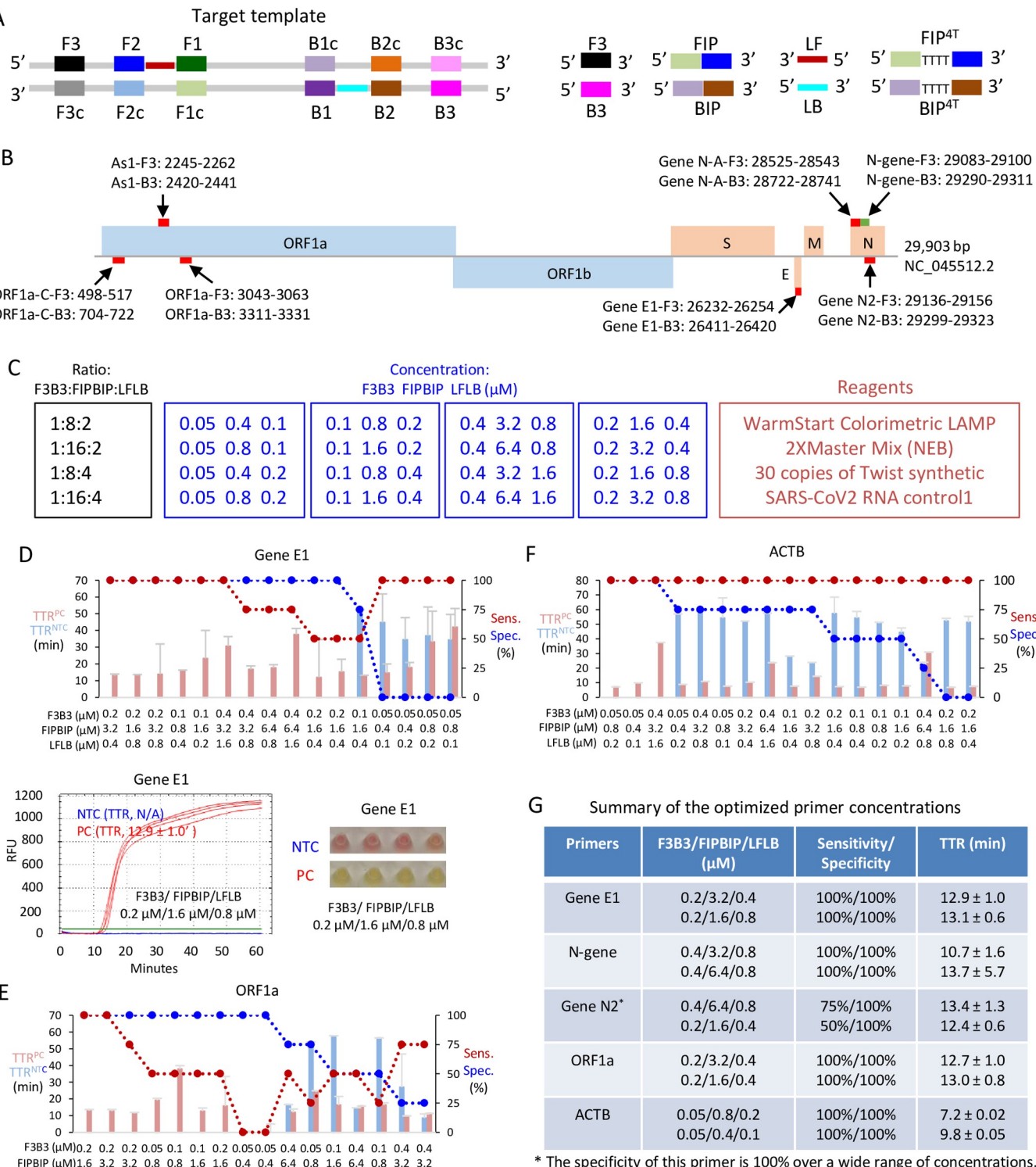

**Fig 1. Screening primer performance at a low copy number of SARS-CoV-2 RNA.** (A) A schematic showing a DNA template amplified by LAMP and the primers targeted to the regions in the template. (B) Location of the 7 target regions for the 14 primer sets in the SARS-CoV-2 genome (NC_045512.2, [34]). The indicated target region is that amplified by the outer F3 and B3 primers. (C) Matrix of test conditions. Each primer set was tested with the indicated primer molar ratio (black), and primer concentrations (blue). A total of 16 conditions were tested for each of the 14 primer sets, with 4 replicates per condition. Other reaction reagents are indicated in red. (D) Screening results for primer Gene E1. Top panel: For the indicated primer mixes (X-axis), red and blue bars indicate TTR using 30 copies of positive control SARS-CoV-2 RNA (TTR^PC) or no template (TTR^NTC), respectively. Red and blue circles indicate sensitivity and

specificity, respectively. Bottom left graph shows an example of the fluorescent signal obtained with STYO 9 dye over the 60 minute reaction period for PC (red) or NTC (blue–undetected) using the indicated Gene E1 primer mix. Green line: Threshold to designate TTR. Bottom right panel shows an example of the phenol red colour at 60 minutes. (E and F) Screening results of primer ORF1a (E) and human ACTB (F); format as in (D). (G) Summary of the best two primer concentrations for the top performing four primer sets with adequate performance based on sensitivity, specificity and TTR. NTC, no template control; PC, positive control (30 copies of SARS-CoV-2 RNA); Sensitivity, the percentage of PC replicates with amplifications; Specificity, the percentage of NTC replicates without amplifications; RFU, relative fluorescence units; TTR, time to results (minutes), the time point that the RFU curve crossing the fluorescent threshold; Error bars represent mean ± standard deviations.

linker between the F1c and F2 as well as B1c and B2 regions of FIP and BIP (Fig 1A) can improve sensitivity [24,33]. Thus, to optimize SARS-CoV-2 detection, we tested 14 primer sets (7 with and 7 without a TTTT insert), which included 12 targeting SARS-CoV-2 and 2 for human β-actin (ACTB, Fig 1B and Table 1). The pilot screen tested 16 different primer concentrations, representing four different primer ratios, with four replicates per test condition (Fig 1C). Accurate estimation of sensitivity and specificity requires more replicates, but we limited the pilot screen to quadruplicates in view of the large survey matrix (14 primers x 16 conditions x 4 replicates = 896 reactions). This approach provided initial approximate sensitivity/specificity estimates to select primers and primer amounts for the next test phase.

To develop a sensitive RT-LAMP assay, we used only 30 copies of synthetic SARS-CoV-2 RNA (Twist Bioscience) in these primer comparisons. To optimize RT-LAMP for ACTB, we used 5ng human RNA because the average RNA concentration of our extracted RNA samples was 5ng/μL and we used 1 ul per RT-LAMP reaction. RT-LAMP reactions were carried out at 65˚C in a thermocycler. 1 μM SYTO 9 green fluorescent dye was used to track the time to result (TTR) in min., deemed as the point at which the RFU (relative fluorescence units) curve crosses the fluorescent threshold (green line, Fig 1D). TTR values were plotted together with sensitivity (% confirmed positives/ positives tested) and specificity (% confirmed negatives/ negative controls tested). The commercial RT-LAMP kit also contains phenol red, which changes from pink to yellow with successful amplification, thus we also recorded color at the end of each experiment. Primer concentration and type affected the specificity of RT-LAMP reactions considerably (Figs 1D–1F and S1), and from this survey the best two primer concentrations of the four top-performing viral primer sets and the best ACTB primer sets were prioritized for Phase 2 (Fig 1G). The selected primer sets included Gene E1, N-gene, Gene N2, and ORF1a, which displayed excellent TTR (~10–14 min), sensitivity (all 100% except Gene N2) and specificity (all 100%). The "TTTT" linker did not improve or impaired performance for 6/7 of the SARS-CoV-2 and the ACTB primers, and although it did improve detection with Gene N-A primers, these remained inferior to the four selected viral primers (S1A Fig). The two best concentrations for ACTB primers were much lower than those of the SARS-CoV-2 primers, and also exhibited a lower TTR than viral primers (Fig 1F and 1G).

## Phase 2: Selection of primers with optimal LoD and specificity

In Phase 2, we increased replicates to 10 (from 4), and assessed 4 (rather than 1) viral template amounts (30, 60, 120, 240 copies). We also assessed four template amounts for ACTB primers (0.01, 0.05, 0.25 or 1.25 ng human RNA). In total, therefore, Phase 2 involved 400 reactions (5 primer sets x 2 primer concentrations x 10 replicates x 4 template concentrations). The LoD is commonly defined as the concentration of analyte that can be detected in 95% of replicates [35], but as we used 10 replicates in this phase, we defined LoD as the lowest copy number at which sensitivity was 100% after 30 min. Specificity was calculated using the no template control (NTC) reactions at both 45 and 60 min. time points, which was used together with the LoD to stratify primer sets (Figs 2A and S2). ORF1a, E1 and N2 primers at the 0.2/3.2/0.4 uM F3B3/FIPBIP/LFLB ratio were the top performers, with LoDs of 120, 240 and 240 copies,

A

| Primers | F3B3/FIPBIP/LFLB (µM) | SARS-CoV2 primers Sensitivity | | | | LoD | Specificity | |
|---|---|---|---|---|---|---|---|---|
| | | (Copies) 30 | 60 | 120 | 240 | | (Min) 45 | 60 |
| ORF1a | 0.2/3.2/0.4 | 4/10 | 9/10 | 10/10 | 10/10 | 120 | 10/10 | 6/10 |
| | 0.2/1.6/0.4 | 3/10 | 5/10 | 8/10 | 10/10 | 240 | 9/10 | 6/10 |
| Gene E1 | 0.2/3.2/0.4 | 4/10 | 9/10 | 9/10 | 10/10 | 240 | 10/10 | 10/10 |
| | 0.2/1.6/0.8 | 3/10 | 6/10 | 9/10 | 10/10 | 240 | 10/10 | 7/10 |
| Gene N2 | 0.2/3.2/0.4 | 2/10 | 4/10 | 8/10 | 10/10 | 240 | 10/10 | 10/10 |
| | 0.2/1.6/0.8 | 6/10 | 6/10 | 9/10 | 10/10 | 240 | 9/10 | 7/10 |
| N-gene | 0.4/3.2/0.8 | 4/10 | 7/10 | 9/10 | 10/10 | 240 | 10/10 | 8/10 |
| | 0.4/6.4/0.8 | 2/10 | 6/10 | 9/10 | 10/10 | 240 | 7/10 | 1/10 |

B

| Primer | F3B3/FIPBIP/LFLB (µM) | ACTB primers Sensitivity | | | | LoD | Specificity | |
|---|---|---|---|---|---|---|---|---|
| | | (ng) 0.01 | 0.05 | 0.25 | 1.25 | | (Min) 45 | 60 |
| ACTB | 0.05/0.4/0.1 | 10/10 | 10/10 | 10/10 | 10/10 | <0.01 | 10/10 | 10/10 |
| | 0.05/0.8/0.2 | 10/10 | 10/10 | 10/10 | 10/10 | <0.01 | 8/10 | 8/10 |

**Fig 2. Evaluation of the optimized primer concentrations based on limit of detection and specificity.** (A) ORF1a, Gene E1, Gene N2 and N-gene primers were assessed at the indicated conditions. Each condition was evaluated with 10 replicates. (B) ACTB primers were evaluated under the indicated conditions. Sensitivity, the percentage of replicates with SARS-CoV-2 RNA or human RNA showing amplifications; Specificity, the percentage of no template controls without amplifications; TTR, time to results (minutes); LoD, limit of detection; Error bars represent mean ± standard deviations.

respectively and 100% specificity at 45 min (Figs 2A and S2D, S2A and S2C). The alternate F3B3/FIPBIP/LFLB ratio for these primers also performed well, but the LoD and/or specificity was marginally weaker (Figs 2A and S2D, S2A and S2C). The N-gene primer set LoDs were similar to the E1 and N2 primers, but specificity was slightly worse at 60 min (Figs 2A and S2B), thus we excluded it from Phase 3. The LoD for the ACTB primers was 100% at all four template concentrations. However, ACTB primer specificity was 100% *vs*. 80% at both 45 and 60 min with the 0.05/0.04/0.1 uM FB3/FIPBIP/LFLB ratio (Figs 2B and S2E), which was thus selected for Phase 3.

## Phase 3: Maximizing sensitivity and specificity with primer multiplexing and supplements

Multiplexed LAMP assays can be used to simultaneously test for multiple pathogens by labeling primers for different pathogens with different fluorophores [36,37]. Here, we tested whether multiplexing the best primer sets from Phase 2 (ORF1a, E1 and N2) improves detection sensitivity of SARS-CoV-2. To reveal differences in sensitivity, we used only 15 copies of viral template, and ran 10 replicates each to compare E1, N2 or ORF1a primer sets alone, or each of the three possible pairings (Figs 3A and S3A). To calculate sensitivity, only fluorescent signals that appeared within 30 mins were counted, whereas specificity of fluorescent NTC reactions were assessed for 60 min. Color reactions were also visually inspected at 60 min. At 15 template copies, the N2 primer sets alone failed, while the E1 or ORF1a primer sets alone

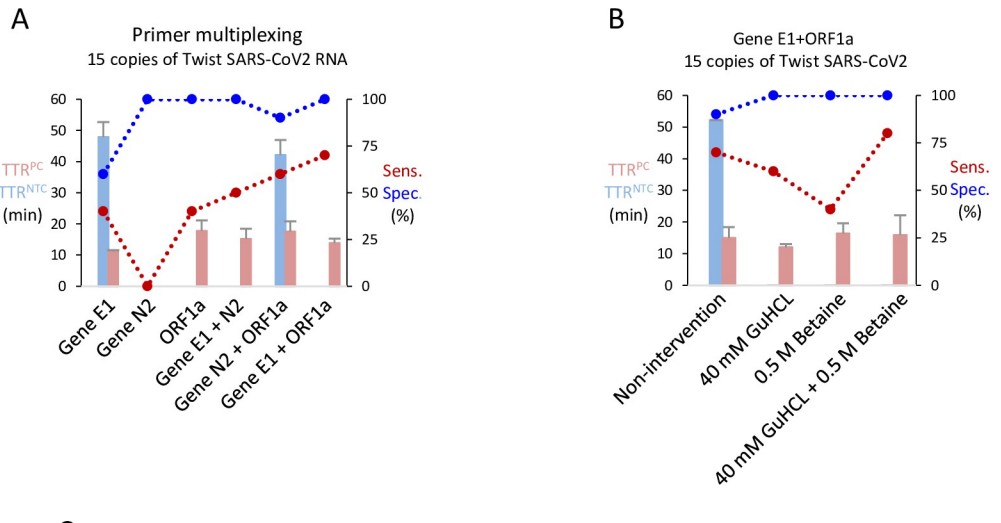

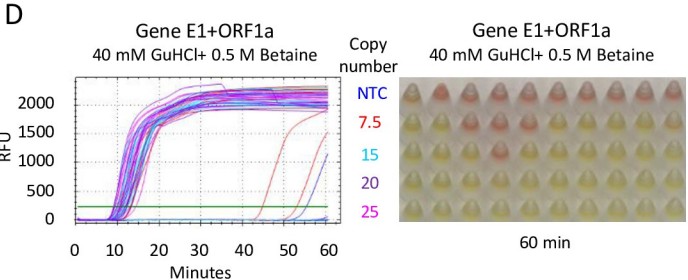

**Fig 3. Effect of primer multiplexing and supplements.** (A) Evaluation of RT-LAMP performance with the indicated primer multiplexing. RT-LAMP reactions were carried out with 15 copies of SARS-CoV-2 RNA at the optimized concentration for each primer set (see Fig 2A in bold). (B) Evaluation of RT-LAMP performance with 40mM GuHCl and/or 0.5M betaine. Reactions were performed with multiplexed Gene E1 and ORF1a primers. (C) LoD assessment of the best RT-LAMP condition with the indicated copy numbers of SARS-CoV-2 RNA. (D) Fluorescent readouts and color changes of the reactions in (C) at 60 minutes. Each condition was evaluated with 10 replicates. NTC, no template control; TTR, time to results; RFU, relative fluorescent units; Error bars represent mean ± standard deviations.

exhibited < 50% sensitivity, and only the latter provided 100% specificity (Figs 3A and S3A). Notably, each of the three primer set pairings improved sensitivity, and the best success rate of 70% success was achieved with the E1 + ORF1a combination, with a mean TTR of < 15 min, and 100% specificity (Fig 3A). At 60 min, 9/10 template reactions (90%) generated the anticipated color change with the E1 + ORF1a combination, while 10/10 NTC reactions remained red (S3B Fig). This dual target primer mix was then taken forward to test whether various additives might further improve performance.

A recent study demonstrated that 40 mM of the denaturing agent guanidine hydrochloride (GuHCl) improves the sensitivity of detection of synthetic SARS-CoV-2 RNA, although patient samples were not tested [26]. A separate study reported that GuHCl did not improve results with patient samples [38]. Betaine, which improves PCR amplification of GC-rich

DNA sequences [39], can also enhance RT-LAMP [40,41]. Thus, we tested whether GuHCl and/or betaine improves sensitivity with the E1 + ORF1a primer combo with 15 copies of viral template. In the unmodified reaction, sensitivity was, as before, 70%, while specificity at 60 min dropped in this experiment from 100% to 90% (*c.f.* Fig 3A *vs.* Figs 3B and S3C), although this aberrant signal appeared beyond 50 min., well after the 30 min. cutoff used to define sensitivity (S3C Fig). Adding GuHCl alone, or more so Betaine alone, reduced sensitivity, but combining GuHCl + Betaine elevated sensitivity to 80%, and in all three of these conditions, specificity was 100% (Figs 3B, S3C and S3D). With this optimized condition, we ran 10 replicates on 4 viral template amounts (7.5, 15, 20, 25 copies), which defined the LoD as 20 copies (2 copies/μl) (Fig 3C and 3D). One out of 10 NTC reactions generated an aberrant signal, but again at beyond 50 min. (Fig 3D). Thus, the combination of E1 and ORF1a primers together with GuHCl and Betaine provided the most sensitive detection of synthetic SARS-CoV-2 RNA.

## Phase 4: Optimized RT-LAMP is comparable to a clinical RT-PCR test with extracted RNA

To test the efficiency of the maximized RT-LAMP in detecting SARS-CoV-2 RNA from clinical patient samples, we tested 30 positive and 36 negative NP samples. RNA was extracted and RT-PCR performed with the BGI RT-PCR kit, which is used in the clinic to detect SARS-CoV-2 ORF1ab and human ACTB [42,43]. Ct values correlated well with clinical Ct values from other detection methods in the positive samples (Fig 4A), and no SARS-CoV-2 was detected in the 36 negative samples (not shown). We then ran RT-LAMP with the Phase-3-optimized conditions and plotted a receiver operator characteristic (ROC) curve of true positive rate (TRP) vs. false positive rate (FPR) ranked on TTRs to evaluate performance. Random assignment of test results generates a diagonal line from 0,0 to 100,100 with an area under the curve (AUC) of 0.5, whereas a perfect test generates vertical line from 0,0 to 0,100 and an AUC of 1.0. RT-LAMP was comparable to the BGI RT-PCR assay, with an AUC of 0.971 (95% CI: 0.896–0.997) (*P* < 0.0001). The TTR at which the TPR + true negative rate (or 1-FPR) is highest was 13.2 minutes, and at that cutoff the sensitivity and specificity of RT-LAMP was 90% (95% CI: 73.5% - 97.9%) and 100% (95% CI: 90.3% - 100%), respectively. The latter satisfies a recommendation from the FDA that tests should exhibit 100% specificity [22]. Of the 3/30 positive samples that were not detectable by RT-LAMP, all were borderline RT-PCR positives with Ct values of 36.0–37.0 (Fig 4C). RT-LAMP successfully detected human ACTB within 20 min in all positive and negative samples (Fig 4D and 4E). Instead of using a thermocycler and a fluorescent readout, we re-ran RT-LAMP with the above 30 positive and 36 negative samples at 65˚C in a water bath for 25 minutes using the end-point colorimetric method, and observed similar sensitivity and specificity (S4A Fig). Thus, with patient-extracted RNA, the optimized RT-LAMP reaction is essentially as sensitive as the gold standard RT-PCR assay used in the clinic, and can be performed using a method (heat source and detection) that is appropriate for low-resource settings, although this would not hold if a high fraction of the population being tested had low copy levels (Ct = 36–37).

To further validate the LAMP assay, it was evaluated in the National Institute of Public Health Research (INSPI) in Ecuador, which employed different clinical protocols for RNA extraction and qRT-PCR to diagnose SARS-CoV-2 infection [28,29]. RT-LAMP was performed with 20 positive and 21 negative NP swab samples. ROC curve analysis indicated an AUC of 1.0 (95% CI: 1.000–1.000) (*P* < 0.0001), and defined the cutoff TTR as 41 minutes. With this cutoff TTR, the assay exhibited 100% sensitivity (95% CI: 83.9% - 100%) and 100% specificity (95% CI: 84.5% - 100%) (Fig 4F and 4G). Human ACTB was detected in all the

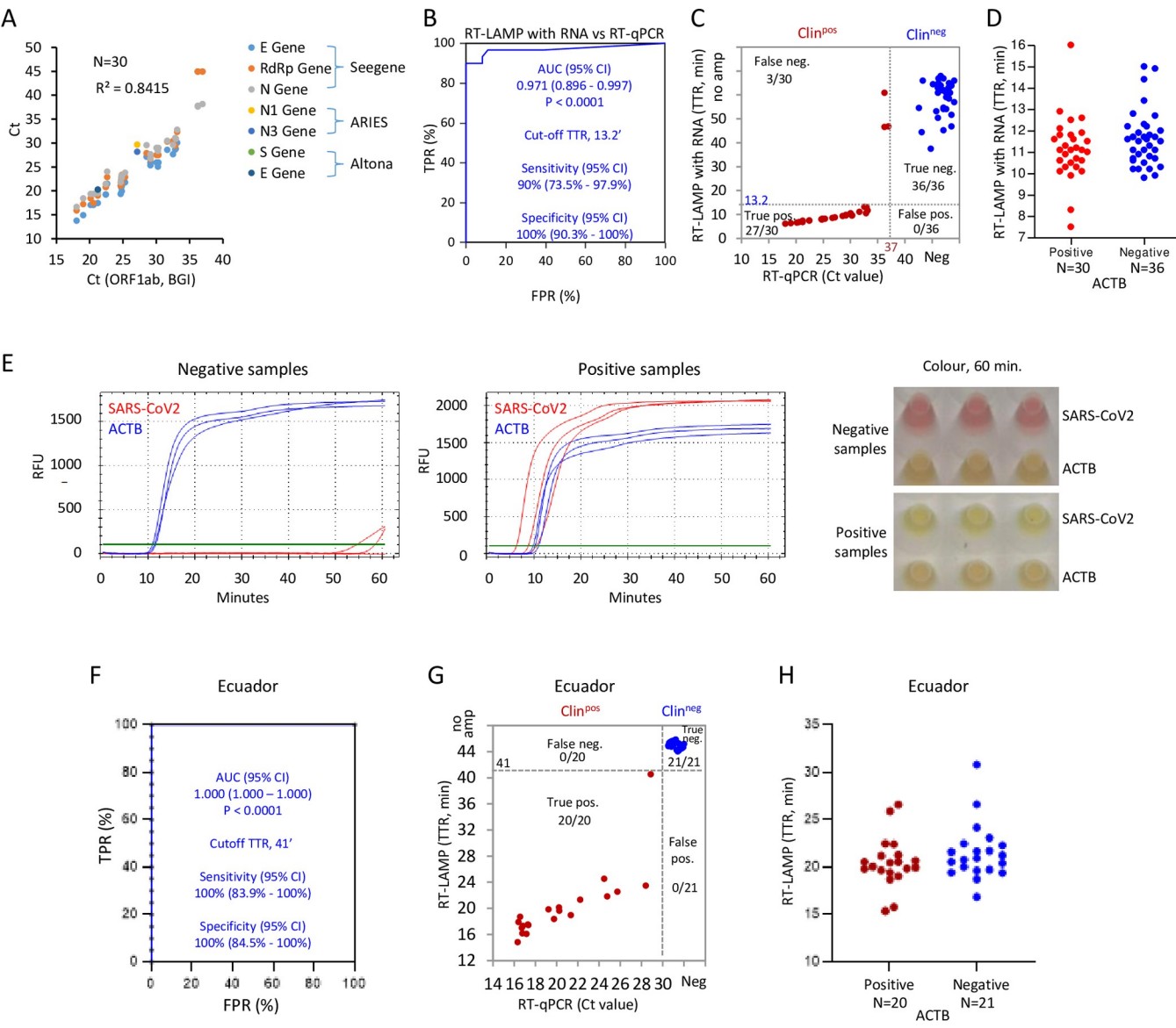

**Fig 4. Comparison of RT-LAMP and BGI RT-PCR with extracted RNA from clinical NP samples.** (A) Correlation of Ct values with BGI RT-PCR kit *vs.* other indicated RT-PCR reagents in 30 SARS-CoV-2 positive clinical NP samples. (B) ROC curve evaluating RT-LAMP performance with 30 positive and 36 negative Canadian clinical samples based on the results of BGI RT-PCR kit. TPR: True positive rate; FPR: False positive rate. TTR ≤ 13.2' was defined as the cut-off to distinguish positive from negative samples with 90% detection sensitivity and 100% specificity. (C) Distribution of RT-LAMP TTRs against BGI RT-PCR Ct values for 30 positive and 36 negative clinical NP samples. BGI RT-PCR and RT-LAMP positives were defined by Ct < 37.0 and TTR ≤ 13.2' respectively. (D) Distribution of human ACTB TTRs. (E) Representative fluorescent readouts and phenol red colour with RT-LAMP reactions at 60 minutes in (C and D) with the clinical NP samples. (F) ROC curve evaluating RT-LAMP performance with 21 positive and 21 negative NP samples from Ecuador. TTR ≤ 41' was defined as the cut-off to distinguish positive from negative samples with 100% specificity and sensitivity. (G) Distribution of RT-LAMP TTRs *vs.* RT-PCR Ct values for gene E with samples in (F). RT-PCR and RT-LAMP positives were defined by Ct ≤ 30.0 for gene E and TTR ≤ 41' respectively. (H) Distribution of human ACTB TTRs of samples in (F).

samples within 35 minutes (Fig 4H). Perhaps reflecting reagent batch differences, TTR in the Ecuadorian lab was longer than that of the Canadian lab for both SARS-CoV-2 and ACTB. Despite this difference, the optimized RT-LAMP performed robustly on extracted patient RNA independent of location or RNA extraction and RT-PCR methods.

## Phase 5: SARS-CoV-2 detection in raw clinical NP samples without RNA extraction

The above tests require access to appropriate resources to purify RNA. Next, therefore, we utilized the clinical NP samples assessed in Phase 4 to determine whether RT-LAMP could be used as a PoC test for direct detection of SARS-CoV-2 without RNA extraction. We utilized the E1 + ORF1a dual primer set and compared amplification with no supplements or the addition of betaine and GuHCl alone or together. 1 μl of raw patient sample in Universal Transport Medium (UTM) was assessed per 10 μl reaction. RT-LAMP successfully detected human ACTB in all samples within 35 min across all the tested conditions, except for Betaine-alone supplementation where the TTRs of one positive and one negative sample were between 40–45 minutes (S5A Fig). To optimize viral RNA detection, we initially assessed four conditions (labeled #2–4 in Fig 5A), which included no supplements, GuHCl alone, Betaine alone, or GuHCl + Betaine and employed ROC curves to identify valid tests. Using a cutoff of $P < 0.001$, recommended for comparing ROC curves [44], only Betaine ($P < 0.0001$) generated an AUC (0.742) that was significantly different from a random test (Fig 5A). Comparing the ROC curves for each condition, the only significant difference was between Betaine alone and GuHCl alone (Fig 5A). These results differed from those obtained with purified RNA, where combining GuHCl and Betaine created a high-performance assay (Figs 3 and 4). These data underscore the importance of optimizing a PoC assay with raw clinical samples.

Although ROC curve analyses confirmed that Betaine supplementation generates a useful test, sensitivity was only 43.3% (Fig 5A). As a fifth condition (labeled #1 in Fig 5A), we modified the Betaine-alone condition by adding 0.25% Igepal CA-630, a detergent that enhances RT-qPCR detection of influenza virus in MDCK cells without RNA extraction [45]. Specificity was 100% in both cases, but sensitivity and AUC increased to 53.3% and 0.771, respectively (Fig 5A). However, this was not significantly different from Betaine alone (sensitivity 43.3%, AUC 0.742), and was still well below the 90% sensitivity and AUC of 0.971 observed with purified RNA (*c.f.* Figs 4B and 5A). Comparing RT-PCR Ct values on purified RNA to RT-LAMP TTR values on raw samples illustrated that the latter performed best on high titer (low Ct) samples (Fig 5B and 5C). Plotting the Ct values of false negatives and true positives with Betaine + Igepal RT-LAMP clarified this bias; 100% (15/15) of samples with Ct ≤ 26.6 were detected, 25% (1/4) sample Ct from 27.1–30 were detected, and no samples (0/11) with Ct > 30 were detected (Fig 5C and 5D). In all samples, human ACTB was detected within 30 minutes (S5A Fig).

To better define sensitivity around the approximate cutoff, we diluted high titer positives with negative patient samples to generate a series of contrived positives with predicted Ct values in the desired range. Direct RT-LAMP detected ACTB in all cases (S5B Fig). Viral RT-LAMP indicated a sensitivity of 100% (95% CI: 67.6%– 100%), 80% (95% CI: 58.4% - 91.9%) and 31.8% (95% CI: 16.4% - 52.7%) for samples with Ct ≤ 25, 25–27.2, and 27.2–29.2, respectively, and all samples with a Ct ≥ 30.0 were false negatives (Fig 5D). In these Canadian samples, Ct values of 25, 27 and 30 corresponded to $7.9 \times 10^6$, $2.5 \times 10^6$ and $3.2 \times 10^5$ copies of SARS-CoV-2 per mL of raw NP samples respectively. Thus, the optimized RT-LAMP assay used directly on 1 μl of NP sample may be a useful screening tool to identify infectious individuals bearing high viral loads [10,46,47], but should not be used to definitively rule out infection.

To test robustness, the assay was evaluated in the diagnostics laboratory, Universidad de Los Andes (Uniandes), Colombia, with 118 positive and 41 negative clinical NP samples. To account for degradation during storage, Ct values were re-assessed with the U-TOP Seasun kit (one-step RT-PCR). To establish a TTR cutoff for use in this setting we assessed 41 negatives

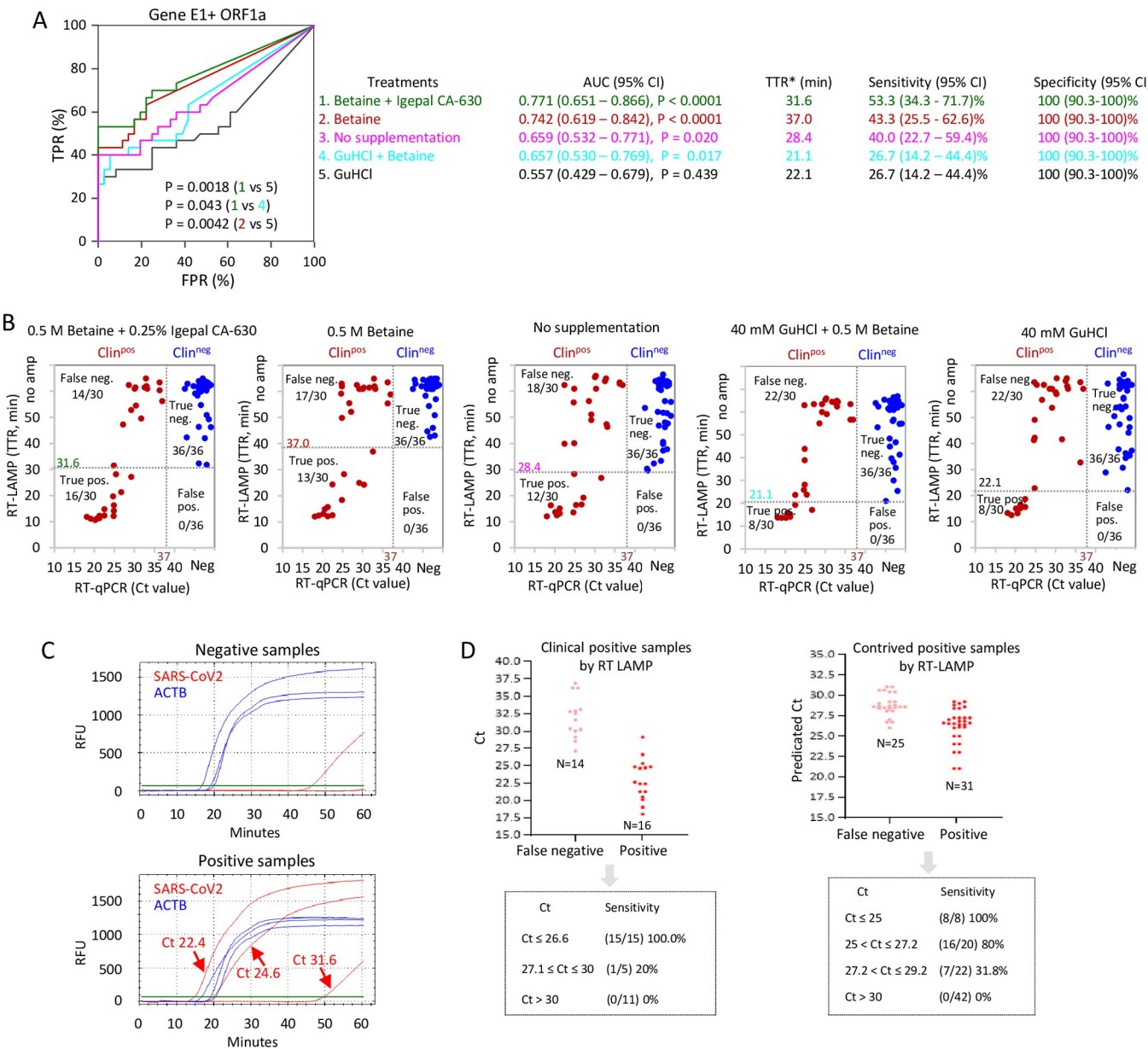

**Fig 5. Direct RT-LAMP on raw clinical NP samples without RNA extraction.** (A) ROC curves evaluating RT-LAMP performance on 30 positive and 36 negative clinical NP samples with the indicated supplements. 0.5M betaine + 0.25% Igepal CA-630 in green; 0.5M betaine in red; No supplements in pink; 40mM GuHCl + 0.5M betaine in light blue; 40mM GuHCl in black. 1μl of raw samples (without any sample processing) was applied to RT-LAMP reactions, and the reactions were carried out with multiplexing primers for Gene E1 and ORF1a. Significance values were calculated with MedCalc software for ROC curve analysis. TTR* indicates the cutoff providing optimal sensitivity and specificity. (B) Distribution of the RT-LAMP TTRs *vs.* BGI RT-PCR Ct values with the indicated supplements. Dotted lines indicate cutoffs. (C) Representative fluorescent readouts of RT-LAMP with 0.5M betaine and 0.25% Igepal CA-630. (D) Sensitivity of RT-LAMP at the indicated Ct ranges. Left panel, Clinical NP samples. Right panel, Contrived positives generated by diluting clinical NP positives with negative NP samples.

and 118 positives, most of which had Ct values < 30. ROC curve analysis generated an AUC of 0.916 ($P$ < 0.0001), and defined the cutoff TTR as 25 minutes with 100% (95% CI: 91.4% - 100%) specificity, recommended by the FDA [22] (Fig 6A). With this cutoff TTR, sensitivity on these selected samples was 49.2% (95% CI: 40.3% - 58.1%) (Fig 6A). As with the Canadian samples, plotting TTR *vs.* Ct showed more efficient detection in high titer (lower Ct) samples

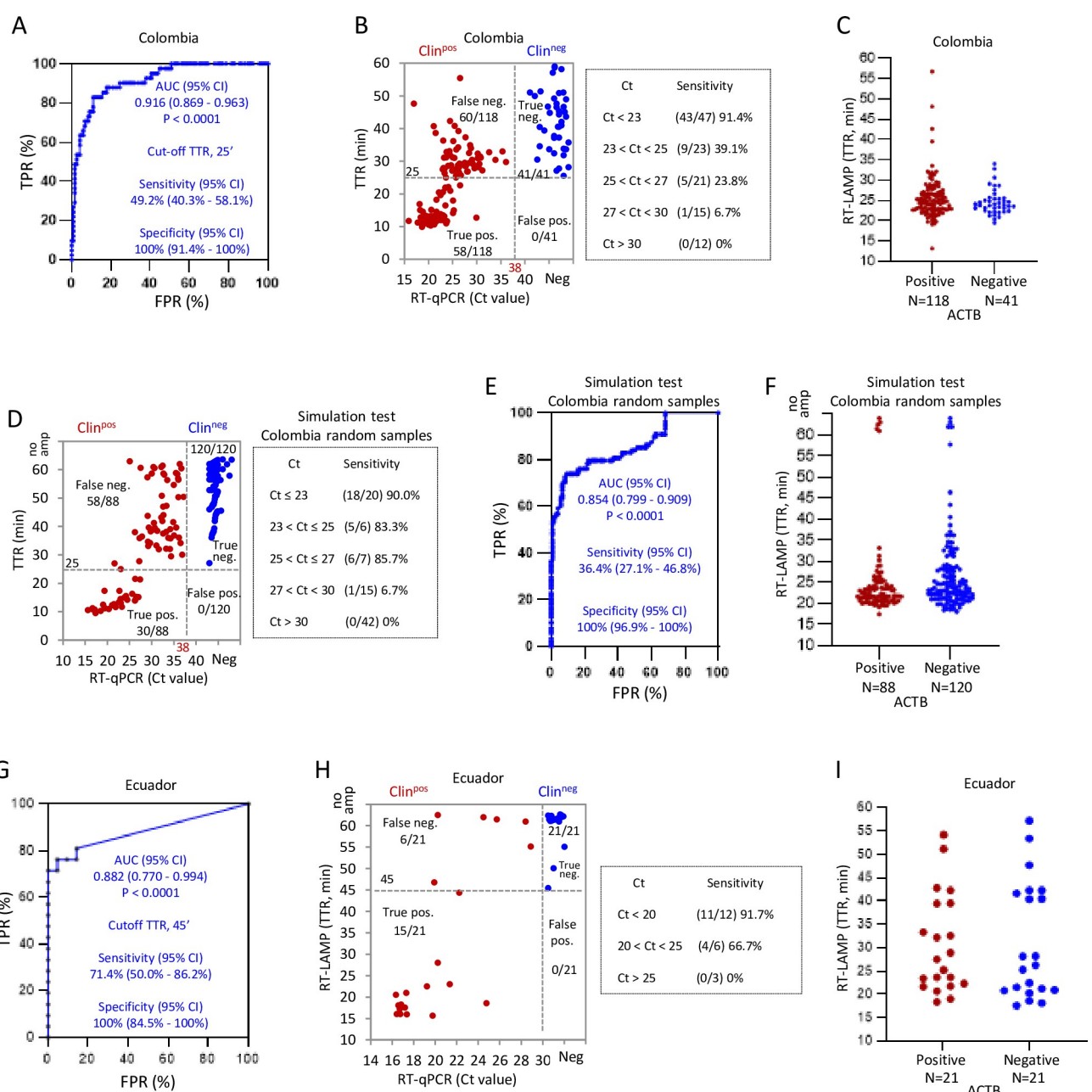

**Fig 6. Direct RT-LAMP on raw clinical NP samples of Colombia and Ecuador.** (A) ROC curve evaluating the optimized direct RT-LAMP performance on 118 positive and 41 negative clinical NP samples from Colombia. (B) Distribution of the RT-LAMP TTRs vs. U-TOP Seasun RT-PCR Ct values for Orf1ab, and the sensitivity of RT-LAMP at the indicated Ct intervals with samples in (A). (C) Distribution of ACTB TTRs of samples in (A). (D) Plot of TTRs *vs.* Ct values in a simulation RT-LAMP test with randomly chosen Colombian samples, and the sensitivity at the indicated Ct intervals with these samples. (E) ROC curve analysis validating the simulation RT-LAMP test in (D). (F) Distribution of ACTB TTRs of samples in (D). (G) ROC curve analysis of the direct RT-LAMP with 21 positive and 21 negative NP samples of Ecuador. (H) Distribution of the direct RT-LAMP TTRs *vs.* RT-PCR Ct values for E gene, and the sensitivity at the indicated Ct values for samples in (G). (I) Distributions of ACTB TTRs of samples in (G). Dotted lines in distribution graphs indicate cutoffs.

(Fig 6B). Sensitivity was 91.4% (95% CI: 80.1%– 96.6%) at Ct < 23, but dropped to 39.1% (95% CI: 22.2%– 59.2%), 23.8% (95% CI: 10.6%– 45.1%), and 6.7% (95% CI: 0.3%– 29.8%) for samples with Ct 23–25, 25–27, and 27–30, respectively, and all samples with a Ct > 30 were false

**Table 3. Comparisions with other RT-LAMP PoC SARS-CoV-2 tests.**

| Studies | Samples | Cts | Specificity | Overall sensitivity | Ct-specific sensitivity | Human gene | Sample pretreatment |
|---|---|---|---|---|---|---|---|
| This study Toronto, Canada | NP samples 30 positive 36 negative | 19.0–36.9 Cutoff ≤ 37 | 100% | 53.3% | 100%, Ct ≤ 26.6 0%, Ct > 30 | ACTB | 56˚C, 30 min* |
| This study Bogota, Colombia Optimization test | NP samples 118 positive 41 negative | 15.9–36.0 Cutoff ≤ 38 | 100% | 49.2% | 91.4%, Ct < 23.0 0%, Ct > 30 | ACTB | None |
| This study Bogota, Colombia Simulation test | NP samples 88 positive 120 negative | 15.6–37.2 Cutoff ≤ 38 | 100% | 36.4% | 90%, Ct ≤ 23.0 0%, Ct > 30 | ACTB | None |
| This study Quito, Ecuador | NP samples 21 positive 21 negative | 16.3–28.9 Cutoff ≤ 30 | 100% | 71.4% | 91.7%, Ct < 20 | ACTB | None |
| This study Toronto, Canada FluoroPLUM | NP samples 30 positive 29 negative | 19.0–36.9 Cutoff ≤ 37 | 100% | 36.7% | 100%, Ct < 22.5 76.9%, Ct < 25 0%, Ct > 30 | ACTB | 56˚C, 30 min* |
| Song et al. [18] | NP samples 19 positive 21 negative | 20–36 | 100% | 84% | 100%, Ct < 32 | None | 56˚C, 1 hour |
| Schermer et al. [19] | NP samples 74 positive 28 negative | 14.3–38.2 | 89.3% | 73% | 97.3%, Ct < 30 | None | 98˚C, 15 min |
| Amaral et al. [20] | Saliva samples 39 positive 15 negative | 18–28 | 100% | 85% | 100%, Ct < 22.2 | None | 95˚C, 30 min |
| Dao Thi et al. [21] | NP samples 128 positive 215 negative | 0–40 | 99.5% | 46.9% | 90.5%, Ct < 25 17.9%, Ct: 30–35 | None | 95˚C, 5 min |
| Papadakis et al. [22] | NP samples 96 positive 67 negative | 8–34 | 100% | 83.3% | 100%, Ct ≤ 25 53.1%, Ct: 30–34 | None | Pretreated with neutralizing buffer |

*, positive samples had been treated for viral inactivation before, not for the purpose of assay optimization.

negatives (Fig 6B). The assay detected ACTB in all 159 samples within 40 minutes except 3 positive samples (Fig 6C).

Using the above TTR cutoff, a simulation test of direct RT-LAMP was then performed with 208 randomly chosen samples (88 positive, 120 negative). We observed 100% specificity (95% CI: 96.9% - 100%) at the preselected TTR cutoff, while sensitivity was again dependent on Ct values, varying from 90.0% (95% CI: 69.9%– 98.2%), 83.3% (95% CI: 43.6%– 99.1%), 85.7% (95% CI: 48.7%– 99.3%), for samples with Ct ≤ 23, 23–25, and 25–27 respectively, but dropping to 6.7% (95% CI: 0.3%– 29.8%) when Ct was 27–30, and all samples with a Ct > 30 were false negatives (Fig 6D). Overall, sensitivity in this setting with these randomly selected samples was only 36.4%, below the 53.3% observed in the Canadian lab (Table 3). The performance of the assay in this simulation test was validated by ROC curve analysis with an AUC 0.854 ($P < 0.0001$) (Fig 6E). Human ACTB was detected in all 208 samples within 40 minutes except for 5 positive and 8 negative samples (Fig 6F).

Finally, direct RT-LAMP was also tested in INSPI laboratory in Ecuador with 21 positive (all reconfirmed Ct values ≤ 30 for Gene E) and 21 negative NP samples. ROC curve analysis confirmed performance with an AUC 0.882 (95% CI: 0.770–0.994) ($P < 0.0001$), sensitivity 71.4% (95% CI: 50.0% - 86.2%) and specificity 100% (95% CI: 84.5% - 100%) at the TTR cutoff defined in this setting of 45 minutes (Fig 6G). As with the Canadian and Colombian data sets,

sensitivity was higher in samples with low Ct values (Fig 6H). ACTB was detected within 45 minutes in most of the samples (Fig 6I). In summary, the overall sample sensitivity of this optimized direct RT-LAMP supplemented betaine and Igepal CA-630 was 53% (16/30), 49.2% (58/118) and 71.4% (15/21) for Canadian, Colombian and Ecuadorian samples respectively while the Ct-specific sensitivity was 100% (15/15) for Ct $\leq$ 26.6, 91.4% (43/47) for Ct < 23 and 91.7% (11/12) for Ct < 20 respectively. Together, these multi-centre studies suggest that this direct RT-LAMP assay has utility as a PoC test only to screen contagious individuals with high viral loads to limit transmission. These data also highlight the real-world fluctuations in sensitivity associated with distinct detection platforms in different locations.

## Phase 6: Direct RT-LAMP with a PoC device: FluoroPLUM

The above direct RT-LAMP protocol removes the need for RNA extraction, but requires a thermocycler. Thus, to further aid PoC testing, we tested a low cost combined incubator and plate reader, FluoroPLUM, developed by LSK Technologies Inc. It is portable and can be operated on any global power supply using the correct plug adaptor, or a portable 12V 10A battery (8–9 h), making it ideal for PoC testing. This device incubates the reaction chamber up to 65˚C and utilizes royal blue LEDs (Luxeon, 440 nm ~ 455 nm), a long pass filter with 515 nm cutoff, and a camera to track change in green channel fluorescence intensity of DNA-bound SYTO 9. Once a 96-well sample plate is loaded on the tray, the device automatically detects wells of interest from captured images and monitors the reaction for 50 minutes. Based on a digital map of the multiwell plate used for the experiments, PLUM software automatically displays graphed results on the screen at the end of the assay (Fig 7A and 7B). Fig 7B highlights the negative sample N1V5 (green font) and the positive sample N1A5 (red font) and the clear visual difference in signal, which is quantified over time in Fig 7A (which plots fluorescence over time for many samples). To interpret the results in FluoroPLUM, we used linear regression to measure "slope$_{20-40}$" (Fig 7C), as it provided easier differentiation among positive and negative samples compared to TTR used in thermocyclers. Slope$_{20-40}$ is calculated using all the data points between 20–40 mins, during which amplification typically occurs in direct RT-LAMP (Figs 5A–5C and 7A). Fig 7C demonstrates the increased signal only in the positive sample. An increase in the concentration of SYTO 9 (10μM) performed better in FluoroPLUM reactions than lower concentrations (S6 Fig and Table 2) and, accordingly, was used in all subsequent assays. From the same group of raw Canadian clinical NP samples as before, we determined slope$_{20-40}$ for 30 positive and 29 negative samples and ran ROC analysis. FluoroPLUM generated an AUC of 0.79 (*P* < 0.0001), comparable to data obtained with a thermocycler (*c.f.* Figs 5A and 7D). The slope$_{20-40}$ at which sensitivity + specificity is highest was 0.0004, and at that cutoff the sensitivity and specificity of RT-LAMP was 70% and 86%, respectively. The latter does not meet FDA guidelines of 100% specificity regarding SARS-CoV-2 molecular test development [22], so we used slope$_{20-40}$ > 0.0048 to define positives, as no false positives were detected above this cutoff. At this cutoff, sensitivity was 36.7% (95% CI: 21.9% to 54.5%), and specificity was 100% (95% CI: 88.3% to 100%). Sensitivity was 100% (95% CI: 67.6% - 100%) or 76.9% (95% CI: 49.7% - 91.8%) for samples with Ct < 22.5 or Ct < 25 respectively, and fell to only 6.3% at Ct > 25 (Fig 7E). In these raw samples, Ct at 22.5 and 25 reflects viral titer at 5.0 X $10^7$ and 7.9 X $10^6$ copies/mL respectively. These results were consistent with reaction color changes visualized at the end of the experiment (Fig 7G and 7H). Fig 7G is the endpoint data for the real-time data shown in Fig 7E (SARS-coV-2) and 7F (ACTB). The 11 true positives in Fig 7E all have a visible green signal in Fig 7G (P11, P14, P18, P20, P23, P25, P26, P29, P30, P34, P36). Thus, the endpoint images match the quantified data. The assay efficiently detected human ACTB with 98.3% sensitivity (58/59) (Fig 7F and 7H). Thus, the portable

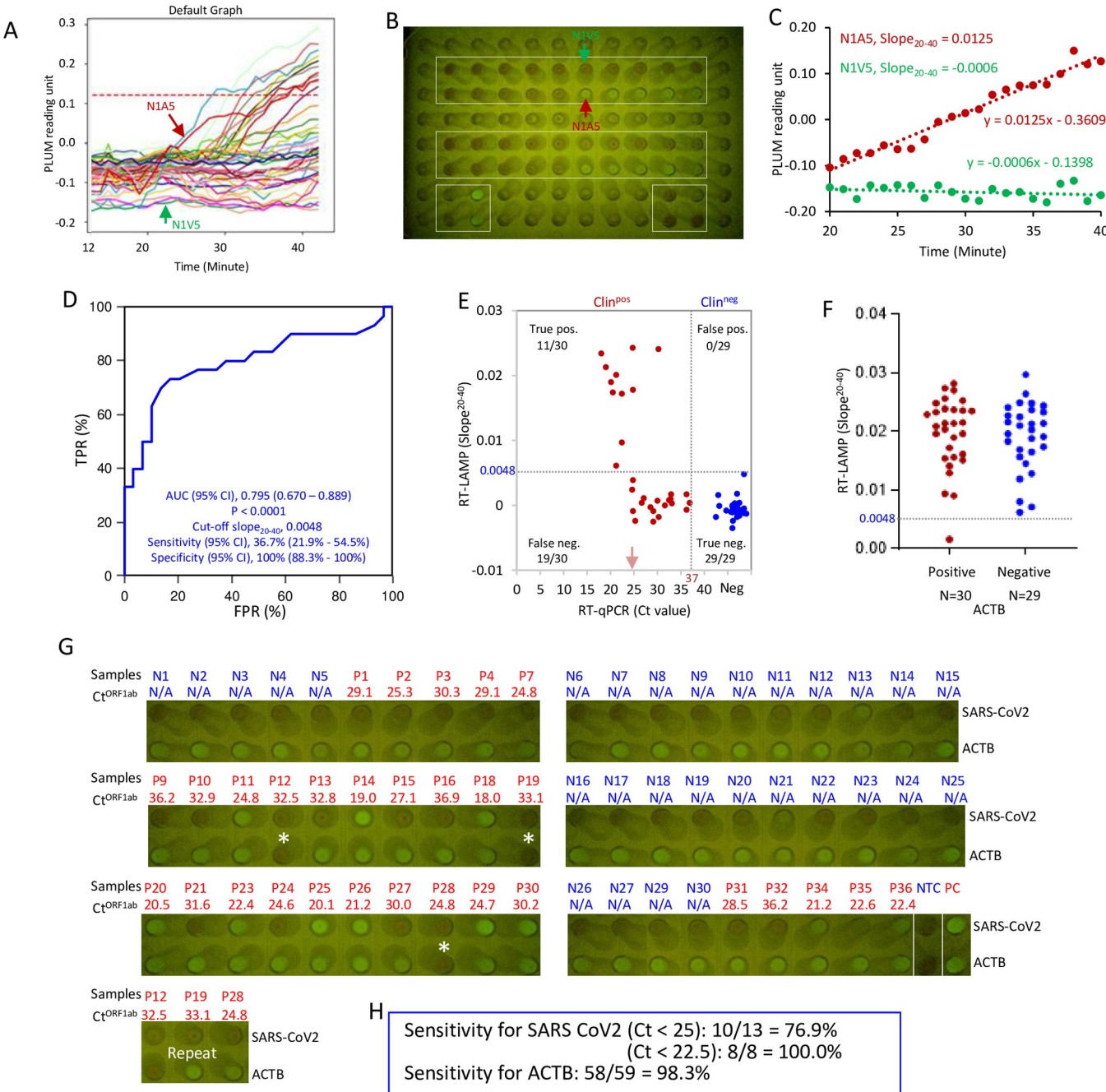

**Fig 7. Direct RT-LAMP with FluoroPLUM.** (A) FluoroPLUM readout of RT-LAMP assessment of the boxed wells in (B). Each solid line represents one reaction, monitored for 45 minutes, and quantified as 'PLUM reading units'. N1A5 (red) and N1V5 (green) are examples of positive and negative samples, respectively. The red dash line is the average reading of all the wells in the plate (B) for the first 3 minutes. (B) Image of RT-LAMP reactions at the end of the experiment. (C) $Slope_{20-40}$ for the two reactions indicated in (A) and (B). (D) ROC curve evaluating FluoroPLUM performance using $slope_{20-40}$ values. (E) Distribution of $slope_{20-40}$ values *vs*. BGI RT-PCR Ct values. Dotted lines indicate cutoffs (RT-LAMP: $Slope_{20-40} > 0.0048$ = positive, RT-PCR: $Ct < 37$ = positive). (F) Distribution of $slope_{20-40}$ for human ACTB in clinical NP samples. (G) Images of color changes for the 30 positive (red) and 29 negative (blue) clinical NP samples. Asterisks: Three samples with no amplification of ACTB, two of which showed amplification in a repeat run (bottom image). (H) Sensitivity with end-point data in (G).

FluoroPLUM instrument, which can be deployed in PoC settings, performs similarly to the thermocyclers used in a diagnostic lab.

## Discussion

Through stepwise optimization of a commercially available RT-LAMP reagent, we developed a SARS-CoV-2 RT-LAMP assay with the potential to be deployed as a PoC test for infectious cases. First, we systemically screened the performance of 7 primer sets (6 for SARS-CoV-2 and 1 for human ACTB), as well as different "TTTT" linker formats, over a wide range of concentrations using a low copy number synthetic SARS-CoV-2 target RNA. Based on sensitivity, specificity and TTR, Gene E1, Gene N2 and ORF1a primer sets were chosen with an ideal concentration for each primer. The ideal concentration for an ACTB primer set was also finalized. We found tremendous variability in performance and ideal concentrations across different primer sets, underscoring the value of the optimization matrix [32]. Multiplexing Gene E1, N2 and/or ORF1a primer sets as well as supplementing reactions with GuHCl and betaine improved sensitivity. The optimized RT-LAMP (multiplexed primers for Gene E1 and ORF1a plus supplementation with 40mM GuHCl and 0.5M betaine) decreased the LoD from 240 copies with the original reagents to 20 copies of SARS-CoV-2 viral RNA per reaction. This improvement has diagnostic significance because each 10-fold increase in the LoD of a COVID-19 viral diagnostic test is expected to increase the false negative rate by 13% [48]. Mapping the optimal primer sets onto sequences of the dominant variants delta and omicron (https://covariants.org/variants) revealed no mismatches. Furthermore, the primer sets target different regions of viral genome, fulfilling an FDA recommendation that molecular tests detect more than one viral genome region [49].

With extracted RNA from clinical NP swab samples, we found the optimized RT-LAMP test was comparable to the BGI RT-qPCR kit, a diagnostic RT-PCR test with top detection sensitivity approved by the FDA [16,43]. ROC curve analysis showed that the AUC was 0.971 ($P < 0.0001$) with 90% sensitivity and 100% specificity. The BGI RT-qPCR protocol defines samples with Ct < 37 for ORF1ab as SARS-CoV-2 positive. RT-LAMP successfully detected SARS-CoV-2 RNA in all positives except three with 36 < Ct < 37, and detected human ACTB in all samples. Notably, the RT-LAMP test takes less than 20 minutes compared with 2 hours for RT-qPCR, and can be performed with a 65°C water bath. For low-resource settings, this reduces the capital investment for RT-qPCR infrastructure (~$25,000), has the potential to bring high fidelity molecular diagnostics to distributed community testing, and reduces the per test cost from ~$25 USD (RT-qPCR) to ~$2.40 (RT-LAMP).

The same assay tested in Ecuador presented an AUC 1.0 with ROC curve analysis, 100% sensitivity and specificity, and detected human ACTB in all the samples. The assay, however, took about 40 minutes, which may reflect differences in reagent sources, sample handling, as well as instrument models. Thus it is important to optimize the cutoff TTR based on different testing conditions. Nonetheless, the strong performance with extracted RNA samples in different countries suggests that RT-LAMP could be deployed when RT-qPCR is limited because of a lack of reagents and/or thermocyclers. Indeed, the WHO considers diagnostic tests with sensitivity $\geq$ 80% and specificity $\geq$ 97% as suitable replacements for laboratory-based RT-PCR if the latter cannot be delivered in a timely manner [50].

Although RT-PCR with purified RNA is the gold standard to confirm SARS-CoV-2 infection, a major limitation is its long turnaround time, especially outside of larger urban centers, compromising test efficacy in terms of timely self-isolation and contact tracing. Rapid and economical PoC tests for SARS-CoV-2, together with masking and social distancing, are necessary to stop community transmission of the disease [1,4]. With this in mind and recognizing that

minimum sample manipulation is essential for PoC tests [14], we next optimized RT-LAMP for raw Canadian NP swab samples without RNA extraction. The best condition used multiplexed Gene E1 and ORF1a primers and supplementation with 0.5M betaine and 0.25% Igepal CA-630. In < 32 mins, the optimized RT-LAMP detected samples with Ct ≤ 25 (viral load ≥ 7.9 X $10^6$ copies/mL) with 100% sensitivity, samples with 25 < Ct ≤ 27.2 (viral load: 2.6 X $10^6$–7.9 X $10^6$ copies/mL) with 80% sensitivity and samples with 27.2 < Ct ≤ 29.2 (viral load: 5.0 X $10^5$–2.0 X $10^6$ copies/mL) with 31.8%. However, this direct test failed to detect SARS-CoV-2 in samples with Ct ≥ 30 (viral load ≤ 3.2 X $10^5$ copies/mL). For all the samples, the RT-LAMP detected human ACTB in less than 30 minutes. Detection of high titer samples was also demonstrated in labs in Colombia and Ecuador. In the former, an initial survey of 41 negative and 118 positive selected samples defined a cutoff TTR of 25 minutes, then a simulation detection test with 88 positive and 120 negative randomly chosen samples displayed 100% specificity, and 90% sensitivity with high titer (Ct ≤ 23) samples. Overall, sensitivity in the Bogota study was 33.4%, below the 55% seen in the Toronto lab. Human ACTB was detected in more than 95% of the samples within 40 minutes. A smaller test in Ecuador with 21 positive (Ct < 30) and 21 negative samples indicated a cutoff TTR of 45 minutes, longer than those of the Canadian and Colombian labs. These results underscore the importance of optimizing RT-LAMP in different locations. Variability may arise from changes in reagents, the logistics of sourcing reagents (e.g. international shipping, time in customs), equipment and personnel, and/or heat-inactivation of clinical samples (Table 3). However, taken together, RT-LAMP brings the potential for deploying molecular testing broadly and, even with a detection threshold limit of Ct 27, could provide significant gains for public health efforts to contain infection. fection.

We compared our direct detection results to those of five other RT-LAMP studies (Table 3). Dao Thi *et al* [20], based in Heidelberg Germany, used N-A gene primers in RT-LAMP, and observed sensitivity of ~47% at 99.5% specificity, which is in a similar range to each of our three cohorts (Table 3). Similar to our work, Schermer *et al*, based in Cologne Germany, developed a multiplex reaction that included guanidine [18]. They reported a sensitivity of 73% with randomly selected samples, above our best result of 53.3%, but specificity was only 89% in contrast to 100% in our study (Table 3). It would be interesting to run comparisons on the same samples with primer sets used in both studies. Song *et al*, using samples from Pennsylvania USA, used a two-step tube reaction ("Penn-RAMP") in which recombinase polymerase amplification (RT-RPA) was performed first in the lid, then after spin-down, RT-LAMP in the tube [17]. They reported 84% sensitivity at 100% specificity, and detected all samples with Ct < 32. Thus, adding the RT-RPA step greatly enhances sensitivity. A drawback is that this strategy requires additional reagents and a centrifuge, which can pose a challenge for operation in low-resource settings. For example, as we experienced, there are no direct suppliers of RPA or LAMP reagents in Colombia or Ecuador, and orders can take 2–6 months to arrive. Moreover, we found the performance of some products was lower than what was experienced in Canada, likely due to disruption of the cold chain during transport or customs clearance. Nation-specific bureaucratic requirements can further delay reagent delivery; for example, the National Institute for Food and Drug Surveillance (INVIMA) in Colombia must approve all reagents, which can affect preservation of reagents requiring cold chain. All of these challenges have been exasperated in the pandemic with, for example, customs personnel working from home and slower administrative approval processes.

It is worth noting that Song *et al* also tested Penn-RAMP with virion RNA in a PoC heating block [17], but whether this approach works with raw samples to the extent seen in the dual-step tube format was not reported. Nevertheless, their data highlight the potential of using RPA to improve RT-LAMP. A study by Papadakis *et al* from Heraklion Greece developed a

portable biomedical device for performing real-time quantitative colorimetric LAMP. They performed RT-LAMP with Bst DNA/RNA polymerase from SBS Genetech, and tested 67 negative and 96 positive crude NP samples [21]. They reported 100% sensitivity at Ct ≤ 25 and 53.1% sensitivity at Ct: 30–34 with 100% specificity. Compared with our results, their reported higher sensitivity at high Ct values is likely due to the polymerase used, which is extremely thermostable and also provides sensitive reverse transcriptase activity. In their study, samples were pretreated with neutralization buffer. Finally, Amaral *et al* [19], based in Lisbon Portugal, assessed saliva rather than NP samples, which has the advantage of easier collection. They observed 85% sensitivity at 100% specificity, but the highest RT-PCR Ct value of their samples was only 28; indeed 100% sensitivity was only observed at Ct < 22.2 (Table 3). Overall, RT-LAMP alone seems best suited to detect high titer samples.

In the final phase of our diagnostic development program, we prototyped deployment of the assay with a portable "lab-in-a-box" that provided combined incubation, optical monitoring and graphing. Direct RT-LAMP with the FluorPLUM device displayed high sensitivity with high viral loads (76.9% for Ct < 25 and 100% for Ct < 22.5), which dropped dramatically with low viral loads (Ct > 25). Thus, most samples with viral load beyond 7.9 X $10^6$ copies/mL could be detected. These data were comparable to results obtained with thermocyclers, justifying future work to assess this strategy in the field. Work is on-going to provide a more robust platform for field testing to broaden accessibility of FluoroPLUM as a PoC device.

Through the lens of maintaining public health, the priority is not necessarily to determine whether a person has any evidence of SARS-CoV-2, but to quickly and accurately identify individuals who are infectious [1]. Various studies have shown that COVID-19 patient sample infectivity correlates with Ct values, and the infectious period corresponds to the period during which viral load is likely to be highest [10–12,51]. Furthermore, a recent study directly demonstrated that viral load of COVID-19 patients was a leading driver of SARS-CoV-2 transmission [52]. In that study, 282 COVID-19 cases were tracked and only 32% led to transmission [52]. Among the 753 total contacts from these cases, the secondary attack rate overall was 17%. Critically, at the lower viral load ($10^6$ copies per mL) the secondary attack rate was 12% compared to 24% when the case had a viral load of $10^{10}$ copies per mL or higher [52]. In comparison with our RT-LAMP assay, the detection sensitivity for Canadian samples with Ct < 25 (viral load > 7.9 X $10^6$ copies per mL) was 76.9%, and 100% for samples with Ct ≤ 22.5 (viral load ≥ 5 X $10^7$ copies per mL). RT-LAMP detected 91.4% of the Colombian samples with Ct < 23, and 91.7% of the Ecuadorian samples with Ct < 20. Ct values were measured with the U-TOP$^{TM}$ COVID-19 detection kit in Colombia, and the SuperScript$^{TM}$ III Platinum$^{TM}$ One-Step RT-PCR System in Ecuador, and Ct < 23 or < 20 corresponds to viral loads of > $10^7$ or > 8 X $10^6$ copies per mL [53,54], respectively. Thus, most individuals with high risk for transmission (> $10^{10}$) would be identified with direct RT-LAMP in all three settings. The turn-around time for laboratory-based RT-PCR testing is generally 24 to 48 hours, and longer in remote areas due to the transport of samples, while the RT-LAMP assay would generate results on-site in less than one hour. Thus, our assay could potentially be deployed as a PoC test at distributed sample collection centers to identify individuals with high risk for transmission to mitigate virus spreading.

A limitation of our study is that we tested NP swab samples processed in research laboratories, not in the field. Also, different RNA extraction methods and RT-PCR kits were used among the three groups. Nevertheless, the lab studies in Colombia and Ecuador indicate feasibility and set the stage for PoC tests. A limitation of all LAMP protocols is that both the supply and cold chains remain a major hurdle for low/mid income nations. Many countries lack the domestic capacity for diagnostic manufacturing and must import health care tools, which, in addition to possible delays and cost, can complicate the response to public health crises. Cell-

free protein expression systems, produced from *E. coli*, offer an exciting solution, and indeed have been applied recently in Chile for detection of a plant pathogen [55]. In addition, extracts of E. coli expressing Bst-LF, which can support LAMP, have been applied recently to detect SARS-CoV2 in reactions that also employ sequence-specific fluorogenic oligonucleotide strand exchange (OSD) probes to minimize false positives [56]. This trend toward locally produced reagents promises to transform diagnostics and reduce costs by orders of magnitude. As a part of the ongoing collaboration among the laboratories included in this study, a research project to locally produce easy-to-implement low-cost kits for the molecular diagnosis of febrile diseases (Sars-Cov-2 and arbovirus), was recently funded by the Ministry of Sciences (Minciencias) in Colombia.

In summary, we developed a rapid RT-LAMP assay for SARS-CoV-2 detection which was essentially as accurate as the BGI RT-PCR kit with extracted RNA, suggesting that it can substitute for laboratory RT-PCR testing. With raw NP samples, the direct RT-LAMP assay detected samples with high viral loads, positioning the assay well for future deployment as a PoC test to control virus spread.

## Supporting information

**S1 Fig. Primer set performance at a low copy number of SARS-CoV-2 RNA (related to Fig 1).** Primer sets for: (A) GeneN-A and GeneN-A$^{4T}$; (B) N-gene and N-gene$^{4T}$; (C) ORF1a-C and ORF1a-C$^{4T}$; (D) Gene E1$^{4T}$ and ORF1a$^{4T}$; (E) Gene N2 and Gene N2$^{4T}$; (F) As1e and ACTB$^{4T}$. NTC, no template control; PC, positive control (30 copies of SARS-CoV-2 RNA); TTR, time to results (min); Error bars represent mean ± standard deviations.
(TIF)

**S2 Fig. Fluorescence plots and end-point color changes for LoD and specificity assays (related to Fig 2).** Fluorescent readouts and images of color changes at 60 minutes are shown for the following primer sets: (A) Gene E1. (B) N-gene. (C) Gene N2. (D) ORF1a. (E) ACTB. Primer amounts (F3B3/FIPBIP/LFBF, μM) are indicated above each assay. Each condition was evaluated with 10 replicates. RFU, relative fluorescence units; NTC, no template control.
(TIF)

**S3 Fig. Effect of primer set multiplexing and guanidine hydrochloride and betaine supplements (related to Fig 3).** (A) Fluorescent readouts of RT-LAMP with the indicated primer multiplexing (also see Fig 3A). (B) Phenol red colour at 60 minutes from assays in (A). (C) Fluorescent readouts of RT-LAMP with multiplexed primer sets for Gene E1 and ORF1a with the indicated supplements (also see Fig 3B). (D) Phenol red colour at 60 minutes from assays in (C). RT-LAMP reactions were performed with 15 copies of SARS-Cov-2 RNA, and each condition was evaluated with 10 replicates. NTC, no template control; RFU, relative fluorescence units.
(TIF)

**S4 Fig. End-point color changes for RT-LAMP reactions with extracted clinical RNA compared to Ct values for RT-PCR (related to Fig 4).** (A) Phenol red colour of RT-LAMP assays with extracted RNA from clinical NP samples. RT-LAMP was carried out in a water batch at 65˚C for 25 minutes with multiplexed Gene E1 and ORF1a primers and 40mM and 0.5M betaine. (B) Sensitivity and specificity in (A). NTC, no template control; PC, positive control (240 copies of SARS-CoV-2 and 1ng human RNA).
(TIF)

**S5 Fig. ACTB detection using the direct RT-LAMP method without RNA extraction (related to Fig 5).** (A) Distribution of ACTB TTRs of raw clinical NP samples under the indicated RT-LAMP conditions. (B) Distribution of ACTB TTRs between RT-LAMP test positive and negative from contrived raw positive NP samples.
(TIF)

**S6 Fig. Optimization of SYTO 9 concentration for direct RT-LAMP with FluoroPLUM (related to Fig 7).** RT-LAMP for ACTB was performed with 10 clinical NP samples with the indicated SYTO 9 concentrations. Bars represented the mean slope$_{20-40}$. A paired t-test was used to assess differences in the means. $^{*}$, $P < 0.05$; $^{**}$, $P < 0.01$.
(TIF)

## Author Contributions

**Conceptualization:** Suying Lu, Rod Bremner.

**Data curation:** Suying Lu, David Duplat, Paula Benitez-Bolivar, Cielo León, Stephany D. Villota, Eliana Veloz-Villavicencio, Valentina Arévalo, Patricio Ponce.

**Formal analysis:** Suying Lu.

**Funding acquisition:** Keith Pardee, Camila González, Rod Bremner.

**Investigation:** Suying Lu, David Duplat, Paula Benitez-Bolivar, Cielo León, Stephany D. Villota, Eliana Veloz-Villavicencio, Valentina Arévalo, Katariina Jaenes, Yuxiu Guo, Seray Cicek, Lucas Robinson, Philippos Peidis, Joel D. Pearson, Patricio Ponce, Silvia Restrepo, John M. González, Adriana Bernal, Marcela Guevara-Suarez.

**Methodology:** Suying Lu, Yuxiu Guo, Seray Cicek.

**Project administration:** Keith Pardee, Varsovia E. Cevallos, Camila González, Rod Bremner.

**Resources:** Jim Woodgett, Tony Mazzulli.

**Software:** Yuxiu Guo, Seray Cicek.

**Supervision:** Keith Pardee, Varsovia E. Cevallos, Camila González, Rod Bremner.

**Validation:** Suying Lu, David Duplat, Stephany D. Villota, Katariina Jaenes.

**Writing – original draft:** Suying Lu, Rod Bremner.

**Writing – review & editing:** Keith Pardee, Varsovia E. Cevallos, Camila González, Rod Bremner.

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
