## [Decision Letter · Decision Letter 0]

21 Mar 2022

PONE-D-22-05378Multicenter international assessment of a SARS-CoV-2 RT-LAMP test for point of care clinical applicationPLOS ONE

Dear Dr. Bremner,

Thank you for submitting your manuscript to PLOS ONE. After careful consideration, we feel that it has merit but does not fully meet PLOS ONE’s publication criteria as it currently stands. Therefore, we invite you to submit a revised version of the manuscript that addresses the points raised during the review process.

We look forward to receiving your revised manuscript.

Kind regards,

Ruslan Kalendar

Academic Editor

PLOS ONE

Journal Requirements:

"I have read the journal's policy and the authors of this manuscript have the following competing interests: Y.G., S.C., and K.P. are co-inventors of the PLUM reader and co-founders of LSK Technologies, Inc."

We note that one or more of the authors are employed by a commercial company: LSK Technologies Inc.

 Reviewers' comments:

Reviewer's Responses to Questions

**Comments to the Author**

1. Is the manuscript technically sound, and do the data support the conclusions?

Reviewer #1: Yes

Reviewer #2: Yes

Reviewer #3: Yes

2. Has the statistical analysis been performed appropriately and rigorously? 

Reviewer #1: Yes

Reviewer #2: Yes

Reviewer #3: Yes

3. Have the authors made all data underlying the findings in their manuscript fully available?

Reviewer #1: Yes

Reviewer #2: Yes

Reviewer #3: Yes

4. Is the manuscript presented in an intelligible fashion and written in standard English?

Reviewer #1: Yes

Reviewer #2: Yes

Reviewer #3: Yes

5. Review Comments to the Author

Reviewer #1: 

The authors reported a study utilizing RT-LAMP for SARS-CoV-2 and human β-40 actin, and tested clinical samples in multiple countries.

There are a few major concerns regarding this paper:

1. The RNA extraction method was not standardized among the samples from Canadian, Columbia and Ecuadorian, It may affect the performance of the RT-LAMP due to differences in RNA quality and quantity.

2. Similarly RT-PCR method performance in different lab may varies due to machine and reagents variations. Targeted gene may produced different outcome as well.

3. Should include methodology of detection limit.

4. Justify: to optimize RT-LAMP for ACTB, what is the reason to use 5ng human RNA as a starting template?

5. Phase 5. Line 405. For RT-LAMP tested using samples without RNA extraction, interested to know how many samples were tested positive by using 1 uL of NP? For those samples that were tested positive, what additives were included?

4. Fig 7. Changes in colour are not significant between positive and negative samples.

5. Author mentioned that direct RT-LAMP can detect NP samples without RNA extraction. What is the limit of detection (maximum CT of sample) of this method?

Reviewer #2: 

This paper describes development and testing of RT-LAMP assays for SARS-CoV-2, using either extracted RNA or crude samples, and using conventional thermocycler or a low-cost portable fluorescence unit. The authors explore several variables including primer ratios/concentrations, additives, and the always-mysterious TTTT linker in the FIP-BIP primers. The authors then test their assay on samples from three sites: Canada, Ecuador, and Colombia.

The results with extracted RNA look very good: as with other RT-LAMP studies, it seems like RT-LAMP does a good job of catching the majority of PCR-positive samples, but becomes sporadic or inconsistent at those samples with low viral load, corresponding to Ct > 35 or so. On the “direct” assays, the results are much less consistent, and the sensitivity (compared to PCR gold standard) drops to a point where the assay is probably not useful and would require further optimization. It’s unclear what direction that ought to take, but it could include some mitigation for RNase degradation, which could be a factor leading to the reduced sensitivity in the direct vs extracted assays.

Some people might consider the poor sensitivity with the direct assay a result not worth publishing, but I find it rather refreshingly honest, and think that it merits publication (especially considering the scope of PLOS One) alongside the (at present) better results with extraction. Overall – the authors present here a large body of work, and while there may be certain things they wish they had done differently at the outset of the study, I think that overall this is a scientifically sound manuscript that will be of general interest to the community developing isothermal amplification tests.

I recommend minor revision to address the following points or questions (note a few of these, don’t actually require revision, but are just things I found interesting as I read through the manuscript).

Colombian samples, line 147-149: please describe criterion for exclusion of invalid samples and/or determination that a sample was degraded. Like, if upon re-analysis, the Ct >= a certain cutoff it was deemed invalid? Is it the criteria that are given in lines 203-211 in the RT-qPCR section?

Optimization of RT-LAMP, lines 172-173: do alcohol and bleach actually do anything to mitigate contamination? Asked differently- what contamination is this mitigating? Live virus or other infectious material? In which case these are reasonable choices. Otherwise- alcohol (ethanol or isopropanol) has no effect on DNA/RNA contamination, and bleach is of questionable effect to mitigate DNA/RNA.

RNA Extraction: It is interesting the 3 sites used 4 different kits/methods for extraction. And notably each site is using a different volume of sample and a different volume for elution. I think since the samples are compared to qPCR and have a Ct to provide a reference for relative amount of RNA, this is ok, although it would have been interesting to know if the extraction method (especially the ratio of sample input to elution volume) has any impact on the RT-LAMP sensitivity.

Contrived samples, line 169: “predicated” Ct values, do they mean “predicted”?

Line 184, Use of photocopier to scan plates -this is an interesting idea.

Line 215: Please define TTR (Time to Result) upon first use. The definition first appears later in the paper (line 257 and then again in line 264).

Figure 1C/1D/1E – One or the other of these is mislabeled with respect to F3B3 and LFLB. I suspect it’s panel D & E, the rows labeled LFLB are actually F3B3, and vice-versa. It is interesting that they arrive at quite a high concentration of FIP/BIP in some cases (1.6 uM appears to be the “standard” in many LAMP publications).

Line 375-377 “Thus, with patient-extracted RNA, the optimized RT-LAMP reaction is essentially as sensitive as the gold standard RT-PCR assay used in the clinic, and can be performed using a method (heat source and detection) that is appropriate for low-resource settings.” – The assumption in this statement is that there will not be many samples in the copy number range (Ct 36-37) where they found RT-LAMP to fail. This is in turn a statement about the patient population, which depends on the intended use, e.g. testing people with symptoms, or non-symptomatic screening, for example. So it is just worth keeping it clear that this statement about sensitivity applies specifically to the patient population from which these samples were drawn.

Figure 4B, C – I am curious about the decision to use TTR <=13.2 min as the cutoff time. From the standpoint of their set of samples, this appears to maximize sensitivity. However, this is a relatively small sample set, and there is quite a lot of time before the earliest false-positives show up (looks like out past 30 minutes). So I would wonder is there a benefit to setting something like TTR <= 20 minutes, in case within a larger set of positive samples there are some positives that amplify a little bit past their 13.2 minutes?

Moving on to the Ecuador samples – they use a different time cutoff. Why does it seem like the LAMP assay is slower on this sample set? Are the LAMP reagents different? (I note this is described much later, in the discussion; perhaps just insert a statement in the results acknowledging the difference?)

Line 635+ - I think the authors devote too much space to discussing & comparing to the RT-RPA + RT-LAMP combination (“Penn RAMP”) – it’s a fundamentally different process.

Line 695-697: using cell-free expression to produce enzymes is interesting. The authors may also want to take a look at the following: https://www.biorxiv.org/content/10.1101/2020.04.13.039941v3 - it’s still a preprint but it looks like it is accepted for publication already. It’s a side point in the paper but they use E. coli cells expressing the enzyme for LAMP as part of the reaction mix.

Reviewer #3: 

In the manuscript, Bremner and coworkers have presented a multicenter international assessment of a SARS-CoV-2 reverse-transcription loop-mediated isothermal amplification (RT-LAMP) test, intended for use in low/medium resource setting. They have optimized the available RT-LAMP, optimized the primers and maximized the sensitivity/specificity of the test, compared the optimized test with clinical RT-PCR test with extracted RNA and more importantly evaluated the RT-LAMP test for SARS-CoV-2 detection in raw clinical naso-pharyngeal samples without RNA extraction in three different research labs in Canada, Colombia and Ecuador. With fresh wave of COVID-19 pandemic rising across the globe, this study on efficient, reliable and cost effective point-of-care testing could not have come at a more important time. The study is well performed and manuscript very well written. A comparison table with other RT-LAMP PoC SARS-CoV-2 tests, specifically stands out as it helps to present the picture quite clearly. Having said this, there are a few concerns that the authors need to address to warrant publication in PLOS ONE.

Major:

• In the discussion (lines 682 : 685), the authors present viral load as a measure for infectivity and detection sensitivity of their optimized RT-LAMP test as 100% for samples with high viral load (Ct ≤ 22.5), and thus the test could be deployed for identification of high risk individuals. However, this is true only for the samples from Canada. In the samples from Colombia and Ecuador, sensitivity and the Ct threshold are varying (Ecuador for example: 90% for Ct ≤ 20). Since these samples would represent better, the intended target use of the developed test, the authors must discuss the corresponding viral loads/infectivity and if indeed the test would be efficient enough for identifying the high-risk individuals. If possible authors could provide some correlation analysis on the viral load/infectivity and their RT-LAMP test sensitivity.

• Have the authors ever tried adding recombinase polymerase amplification (RT-RPA) to their optimized test? It would be interesting to compare with the sensitivity of assay with that of the one developed by Song et al. and hence perhaps a better applicability of their test with RT-RPA.

Minor:

• Any particular reason why the heat inactivation was not carried out in the samples from Colombia and Ecuador? If so, it can be added in the manuscript.

• Please check carefully if the abbreviations are explained the first time they appear in the manuscript. Example: Line 98: FDA not expanded, COVID 19 full form in the abstract

• Many references lack page range. Example: Ref. nos. 21, 34, 46. Please check the references carefully.

6. PLOS authors have the option to publish the peer review history of their article (what does this mean?). If published, this will include your full peer review and any attached files.

Reviewer #1: No

Reviewer #2: No

Reviewer #3: **Yes: **Revathi Sekar

---

## [Author Response · Author response to Decision Letter 0]

20 Apr 2022

Reviewer #1: 

The authors reported a study utilizing RT-LAMP for SARS-CoV-2 and human β-40 actin, and tested clinical samples in multiple countries.

There are a few major concerns regarding this paper:

We thank the reviewer for their thoughtful comments. Their concerns are addressed below.

1. The RNA extraction method was not standardized among the samples from Canadian, Columbia and Ecuadorian, It may affect the performance of the RT-LAMP due to differences in RNA quality and quantity.

The reviewer is correct and we agree that in an ideal world, all methods and reagents would be identical in the three different nations. In reality, this is challenging because the availability of reagents is a major road block in low income nations. Even when items are available from the same company, delivery can be affected by long delays in customs or by other nation-specific regulatory delays for imported goods, all of which can compromise reagent integrity. All nations were able to utilize the NEB Warmstart reagents for the RT-LAMP reactions, although we had to arrange for shipping of LAMP reagents from NEB HQ (Ipswish, MA) to Bogota to improve quality, highlighting the challenges of working in low resource settings. For RNA extraction we decided to utilize the methods that were already in place and working in each nation, and that matched the methods used to perform clinical RT-PCR reactions. We have modified the Discussion of limitations to indicate this important caveat (page 35, line 719 - 720). 

2. Similarly RT-PCR method performance in different lab may varies due to machine and reagents variations. Targeted gene may produced different outcome as well.

The reviewer is also correct re RT-PCR methods, but the same limitations apply as noted above, which also applies to machines used to perform RT-PCR. In each country, we opted to use the local clinical RT-PCR diagnostic on-site as the gold standard for that location. While we acknowledge that this approach is imperfect, this is the most practical solution to instigate new diagnostic methodology, particularly in low-income nations. The adjusted Discussion also highlights this caveat (page 35, line 719 - 720).

3. Should include methodology of detection limit.

In the original manuscript, we integrated detection limit methodology into the ‘Results’ section. As the reviewer recommended, we have now described the methodology in detail in the ‘Materials and Methods’ section (page 10, line 172-179). 

4. Justify: to optimize RT-LAMP for ACTB, what is the reason to use 5ng human RNA as a starting template?

The average concentration of extracted RNA from our clinical NP samples was around 5ng/μL and as we used 1 μL of RNA for RT-LAMP reactions, we used 5ng of human RNA to optimize ACTB primers. As the reviewer recommended, we have now explained rationale in the text (page 15, line 273- 274). 

5. Phase 5. Line 405. For RT-LAMP tested using samples without RNA extraction, interested to know how many samples were tested positive by using 1 uL of NP? For those samples that were tested positive, what additives were included?

As indicated in Fig 5B, in the Canadian samples, RT-LAMP detected 16 out of 30 positive clinical samples; in the Colombian samples it detected 58 out of 118; and in the Ecuadorian samples it detected 15 out of 21. For all these samples, 0.5M betaine and 0.25% Igepal CA-630 were included. We have also included this information in the modified text (page 26, line 520 - 524).

4. Fig 7. Changes in colour are not significant between positive and negative samples.

Fig 7B highlights the negative sample N1V5 (green font) and the positive sample N1A5 (red font) and the clear visual difference in signal, which is quantified over time in Fig 7A (which plots fluorescence over time for many samples) and 7C (which calculates slope), and these graphs demonstrate the increased signal only in the positive sample. 

Fig 7G is the endpoint data for the real-time data shown in Fig 7E (SARS-CoV-2) and 7F (ACTB). As noted in Fig 7E, there are 11/30 clinical positives that were also detected by LAMP (true positives), and 19/30 clinical positives that failed in the LAMP reaction (false negatives). The 11 true positives all have a visible signal in Fig 7G (P11, P14, P18, P20, P23, P25, P26, P29, P30, P34, P36). Fig 7H highlights the fact that the true positives all have low Ct values (e.g. 8/8 samples with Ct < 22.5 were detected by LAMP). Thus, the endpoint images match the quantified data. We have added more detail to the text to clarify the positives as detailed above (page 27, lines 539 – 541; 545 – 546; page 28, lines 560 - 563).

5. Author mentioned that direct RT-LAMP can detect NP samples without RNA extraction. What is the limit of detection (maximum CT of sample) of this method?

To define “LoD” would require 20 repeat assays at multiple concentrations of a sample to establish a level at which at least 19/20 score positive. LoD assays are common with purified RNA (where it is more straightforward to set up standard amounts of RNA) but not with raw samples. Most publications determine the sensitivity and specificity for raw samples. We ran ROC curves to establish which approaches were better than random, then established sensitivity and specificity. With respect to the maximum Ct detected, we reported 100% sensitivity for Ct ≤ 26.6, 91.4% sensitivity for Ct < 23, and 91.7% for Ct < 20 in Canadian, Colombian and Ecuadorian studies, respectively. These data are shown in Figs 5D, 6B and 6H, but in addition, we now also added a summary of this information to the Results section (page 26, line 520 - 524). 

 

Reviewer #2: 

This paper describes development and testing of RT-LAMP assays for SARS-CoV-2, using either extracted RNA or crude samples, and using conventional thermocycler or a low-cost portable fluorescence unit. The authors explore several variables including primer ratios/concentrations, additives, and the always-mysterious TTTT linker in the FIP-BIP primers. The authors then test their assay on samples from three sites: Canada, Ecuador, and Colombia.

The results with extracted RNA look very good: as with other RT-LAMP studies, it seems like RT-LAMP does a good job of catching the majority of PCR-positive samples, but becomes sporadic or inconsistent at those samples with low viral load, corresponding to Ct > 35 or so. On the “direct” assays, the results are much less consistent, and the sensitivity (compared to PCR gold standard) drops to a point where the assay is probably not useful and would require further optimization. It’s unclear what direction that ought to take, but it could include some mitigation for RNase degradation, which could be a factor leading to the reduced sensitivity in the direct vs extracted assays.

Some people might consider the poor sensitivity with the direct assay a result not worth publishing, but I find it rather refreshingly honest, and think that it merits publication (especially considering the scope of PLOS One) alongside the (at present) better results with extraction. Overall – the authors present here a large body of work, and while there may be certain things they wish they had done differently at the outset of the study, I think that overall this is a scientifically sound manuscript that will be of general interest to the community developing isothermal amplification tests.

I recommend minor revision to address the following points or questions (note a few of these, don’t actually require revision, but are just things I found interesting as I read through the manuscript).

We thank the reviewer for their encouraging comments (“refreshingly honest”, “merits publication”; “scientifically sound”). While sensitivity was excellent with pure RNA, it was indeed frustrating that the sensitivity on raw samples was not higher. Nevertheless, those are the real-world results, and as the reviewer graciously noted they will be of general interest to the community. Our work also reflects the challenges of setting up and standardizing point-of-care assays across nations, particularly in low-income settings. Adding modifications (such as RPA, see Discussion) to the procedure, could improve sensitivity, but may generate additional issues with reagent availability. 

Colombian samples, line 147-149: please describe criterion for exclusion of invalid samples and/or determination that a sample was degraded. Like, if upon re-analysis, the Ct >= a certain cutoff it was deemed invalid? Is it the criteria that are given in lines 203-211 in the RT-qPCR section?

We thank the reviewer for pointing out this oversight. Samples with Ct > 38 for Orf1ab, N gene and RNase P were deemed invalid. We have now added this information to the manuscript (page 8, line 144 – 145 and page 9, line 148). 

Optimization of RT-LAMP, lines 172-173: do alcohol and bleach actually do anything to mitigate contamination? Asked differently- what contamination is this mitigating? Live virus or other infectious material? In which case these are reasonable choices. Otherwise- alcohol (ethanol or isopropanol) has no effect on DNA/RNA contamination, and bleach is of questionable effect to mitigate DNA/RNA.

The reviewer is correct that the purpose of alcohol and bleach is mainly for decontamination (of organisms), and we have modified the manuscript accordingly (page 10, line 181). We did try UDG in our optimization assays to prevent contamination with amplified DNA, but found it decreased specificity so we did not continue with that approach.

RNA Extraction: It is interesting the 3 sites used 4 different kits/methods for extraction. And notably each site is using a different volume of sample and a different volume for elution. I think since the samples are compared to qPCR and have a Ct to provide a reference for relative amount of RNA, this is ok, although it would have been interesting to know if the extraction method (especially the ratio of sample input to elution volume) has any impact on the RT-LAMP sensitivity.

For practical purposes, the RNA extraction methods were those already in use in the clinics on-site. We are uncertain the extent to which altering the ratio of sample input to elution volume may alter sensitivity, but since the sensitivity and specificity was very high for extracted RNA in different settings, and the main goal was to test RT-LAMP as a point-of-care method, we did not go back and revalidate the assay with extracted RNA using identical extraction methods. 

Contrived samples, line 169: “predicated” Ct values, do they mean “predicted”?

Thank you for noting that error. It has now been fixed (page 10, line 169).

Line 184, Use of photocopier to scan plates -this is an interesting idea.

Thank you. 

Line 215: Please define TTR (Time to Result) upon first use. The definition first appears later in the paper (line 257 and then again in line 264).

Thank you for noting this mistake. It has now been corrected. The definition of TTR is now on (page 12, line 224). 

Figure 1C/1D/1E – One or the other of these is mislabeled with respect to F3B3 and LFLB. I suspect it’s panel D & E, the rows labeled LFLB are actually F3B3, and vice-versa. It is interesting that they arrive at quite a high concentration of FIP/BIP in some cases (1.6 uM appears to be the “standard” in many LAMP publications).

Thank you for pointing out this mix up. We have corrected accordingly.

Line 375-377 “Thus, with patient-extracted RNA, the optimized RT-LAMP reaction is essentially as sensitive as the gold standard RT-PCR assay used in the clinic, and can be performed using a method (heat source and detection) that is appropriate for low-resource settings.” – The assumption in this statement is that there will not be many samples in the copy number range (Ct 36-37) where they found RT-LAMP to fail. This is in turn a statement about the patient population, which depends on the intended use, e.g. testing people with symptoms, or non-symptomatic screening, for example. So it is just worth keeping it clear that this statement about sensitivity applies specifically to the patient population from which these samples were drawn.

The reviewer is correct. We have added: “although this would not hold if a high fraction of the population being tested had low copy levels (Ct = 36-37).”. (page 20, lines 388 - 389) 

Figure 4B, C – I am curious about the decision to use TTR <=13.2 min as the cutoff time. From the standpoint of their set of samples, this appears to maximize sensitivity. However, this is a relatively small sample set, and there is quite a lot of time before the earliest false-positives show up (looks like out past 30 minutes). So I would wonder is there a benefit to setting something like TTR <= 20 minutes, in case within a larger set of positive samples there are some positives that amplify a little bit past their 13.2 minutes?

The 13.2 min cutoff for TTR is derived from the ROC curve in Fig 4B, but the reviewer is correct that it need not be so rigorous. It’s likely that with a much larger sample size, a new ROC curve would indicate a longer TTR cutoff time. 

Moving on to the Ecuador samples – they use a different time cutoff. Why does it seem like the LAMP assay is slower on this sample set? Are the LAMP reagents different? (I note this is described much later, in the discussion; perhaps just insert a statement in the results acknowledging the difference?)

This longer time is also defined by a ROC curve (Fig 4G). All but one of the positives is detected by RT-LAMP in under 25 min., and there is one outlier at ~40 min. However, because the latter is still below the shortest time for a signal in the clinically negative samples, 41 min ends up being the TTR cutoff defined by the ROC curve. Nevertheless, 25 min is still longer than 13 mins. The LAMP reagents are all from NEB, but may differ in quality due to lot variation and/or delivery/storage issues. As suggested by the reviewer, we added a statement at this point in the Results to acknowledge the difference in TTR cutoff (page 21, lines 415 - 416).

Line 635+ - I think the authors devote too much space to discussing & comparing to the RT-RPA + RT-LAMP combination (“Penn RAMP”) – it’s a fundamentally different process.

The sensitivity was much higher in that approach, so we felt it was valuable to acknowledge that their reaction is better, but also to highlight the caveat that it may be challenging to set up as a PoC assay in low-income settings. We would prefer to keep this Discussion item. 

Line 695-697: using cell-free expression to produce enzymes is interesting. The authors may also want to take a look at the following: https://www.biorxiv.org/content/10.1101/2020.04.13.039941v3 - it’s still a preprint but it looks like it is accepted for publication already. It’s a side point in the paper but they use E. coli cells expressing the enzyme for LAMP as part of the reaction mix.

We thank the reviewer for highlighting this new manuscript. We added the following sentence to the Discussion: “In addition, extracts of E. coli expressing Bst-LF, which can support LAMP, have been recently applied to detect SARS-CoV2 in reactions that also employ sequence-specific fluorogenic oligonucleotide strand exchange (OSD) probes to minimize false positives (page 36, line 726 – 729). 

Reviewer #3: 

In the manuscript, Bremner and coworkers have presented a multicenter international assessment of a SARS-CoV-2 reverse-transcription loop-mediated isothermal amplification (RT-LAMP) test, intended for use in low/medium resource setting. They have optimized the available RT-LAMP, optimized the primers and maximized the sensitivity/specificity of the test, compared the optimized test with clinical RT-PCR test with extracted RNA and more importantly evaluated the RT-LAMP test for SARS-CoV-2 detection in raw clinical naso-pharyngeal samples without RNA extraction in three different research labs in Canada, Colombia and Ecuador. With fresh wave of COVID-19 pandemic rising across the globe, this study on efficient, reliable and cost effective point-of-care testing could not have come at a more important time. The study is well performed and manuscript very well written. A comparison table with other RT-LAMP PoC SARS-CoV-2 tests, specifically stands out as it helps to present the picture quite clearly. Having said this, there are a few concerns that the authors need to address to warrant publication in PLOS ONE.

We thank the reviewer for their encouraging comments (“well-performed” “very well written” etc.). Their comments are addressed below. 

Major:

• In the discussion (lines 682 : 685), the authors present viral load as a measure for infectivity and detection sensitivity of their optimized RT-LAMP test as 100% for samples with high viral load (Ct ≤ 22.5), and thus the test could be deployed for identification of high risk individuals. However, this is true only for the samples from Canada. In the samples from Colombia and Ecuador, sensitivity and the Ct threshold are varying (Ecuador for example: 90% for Ct ≤ 20). Since these samples would represent better, the intended target use of the developed test, the authors must discuss the corresponding viral loads/infectivity and if indeed the test would be efficient enough for identifying the high-risk individuals. If possible authors could provide some correlation analysis on the viral load/infectivity and their RT-LAMP test sensitivity.

The reviewer is correct, and we agree that it is important to estimate the corresponding viral loads/infectivity with direct RT-LAMP detection sensitivity in Colombia and Ecuador. For the Columbian samples, Ct values were measured with U-TOP COVID-19 detection kit. Based on a recent report for comparing three molecular diagnostic assays for SARS-CoV-2 detection (Kim H-N et al.), Ct < 23 corresponds to a viral load > 107 copies per mL. For the Ecuadorian samples, Ct values were determined with SuperScriptTM III PlatinumTM One-Step RT-qPCR System. Based on a recent report using this system as a standard to evaluate other in-house developed enzymes for SARS-CoV-2 detection (Takahashi M et al.), Ct < 20 also corresponds to a viral load > 107copies per mL. Based on a study from Spain (Marks et al.), the majority of SARS-CoV-2 transmission occurs with a viral load of 1010 copies per mL or higher. Thus, this suggests that the RT-LAMP would contribute significantly in identifying the high-risk individuals. As the reviewer recommended, we have now added this valuable information to the manuscript (page 35, line 708 - 713). 

• Have the authors ever tried adding recombinase polymerase amplification (RT-RPA) to their optimized test? It would be interesting to compare with the sensitivity of assay with that of the one developed by Song et al. and hence perhaps a better applicability of their test with RT-RPA.

This is an interesting approach, which we noted in the Discussion. We did not attempt RT-RPA as it would have added a new level of complexity and requirement for additional reagents, which complicates PoC application, particularly in low income settings. We noted these issues in the Discussion, and also emphasized the use of bacterially expressed, dried reagents may help overcome these drawbacks (and we added a new reference in this regard that was highlighted by Reviewer #2). 

Minor:

• Any particular reason why the heat inactivation was not carried out in the samples from Colombia and Ecuador? If so, it can be added in the manuscript.

We used samples as available from the clinics in each location. 

• Please check carefully if the abbreviations are explained the first time they appear in the manuscript. Example: Line 98: FDA not expanded, COVID 19 full form in the abstract

We thank the reviewer for pointing out this oversight. The modified version explains “FDA” on page 5, line 98, and “COVID-19” is explained in the abstract. We also explain TTR at first appearance (as pointed out by Reviewer #2). 

• Many references lack page range. Example: Ref. nos. 21, 34, 46. Please check the references carefully.

We added the page range where it was missing. Thank you for highlighting this problem.

---

## [Editor Report · Decision Letter 1]

28 Apr 2022

Multicenter international assessment of a SARS-CoV-2 RT-LAMP test for point of care clinical application

PONE-D-22-05378R1

Dear Dr. Bremner,

We’re pleased to inform you that your manuscript has been judged scientifically suitable for publication and will be formally accepted for publication once it meets all outstanding technical requirements.

Kind regards,

Ruslan Kalendar

Academic Editor

PLOS ONE

---

## [Editor Report · Acceptance letter]

2 May 2022

PONE-D-22-05378R1 

Multicenter international assessment of a SARS-CoV-2 RT-LAMP test for point of care clinical application 

Dear Dr. Bremner:

I'm pleased to inform you that your manuscript has been deemed suitable for publication in PLOS ONE. Congratulations! Your manuscript is now with our production department. 

Kind regards, 

on behalf of

Professor Ruslan Kalendar 

Academic Editor

PLOS ONE